# Hemispheric differences in ozone across the stratosphere-troposphere exchange region

Rodrigo J. Seguel [1, 2], Charlie Opazo [1, 2], Yann Cohen [3], Owen R. Cooper [4], Laura Gallardo [1, 2], Björn-Martin Sinnhuber [5], Florian Obersteiner [5], Andreas Zahn [5], Peter Hoor [6], Susanne Rohs [7], Andreas Marsing [8]

[1]Center for Climate and Resilience Research (CR)[2], Santiago, Chile
[2]Department of Geophysics, Faculty of Physical and Mathematical Sciences, University of Chile, Santiago, Chile
[3]Institut Pierre-Simon Laplace (IPSL), Sorbonne Université/CNRS, Paris, France
[4]NOAA Chemical Sciences Laboratory, Boulder, USA
[5]Karlsruhe Institute of Technology, Institute of Meteorology and Climate Research (IMK), Karlsruhe, Germany
[6]Institute for Atmospheric Physics, Johannes Gutenberg-University, Mainz, Germany
[7]Institute of Energy and Climate Systems 3 – Troposphere (ICE-3), Forschungszentrum Jülich, Jülich, Germany
[8]Institute of Atmospheric Physics, German Aerospace Center (DLR), Oberpfaffenhofen, Germany

*Correspondence to*: Rodrigo J. Seguel (rodrigoseguel@uchile.cl)

**Abstract**

Ozone changes in the upper troposphere-lower stratosphere (UTLS) resulting from dynamical and chemical processes strongly affect the atmosphere's radiative forcing. This study analyzed intra- and interhemispheric ozone differences in the UTLS within the 45-60° latitude band, distinguishing between years disrupted by sudden stratospheric warming (SSW) events from 2002 to 2022. We utilized measurements from IAGOS commercial aircraft (45-60 ºN), long-term ozonesonde records (45-60 ºS) and unique in situ measurements from the High Altitude and Long Range (HALO) research aircraft deployed over the southernmost region of South America (45-60 ºS) in September-November 2019 as a part of the Southern Hemisphere Transport, Dynamics, and Chemistry (SouthTRAC) research campaign. The mission period enabled us to examine the impact of two Southern Hemisphere (SH) SSW events developed during 2002 and 2019. To enhance the spatial coverage, we incorporated CAMS reanalysis data. Stratospheric air origin was assigned using relative humidity (<20%) and carbon monoxide (<50 nmol mol[-1]) thresholds. Our results show that air masses of stratospheric origin had higher ozone abundances in the Northern Hemisphere (NH) UTLS than in the SH (between 300-200 hPa and 45-60° latitude): In high ozone depletion years in the stratospheric vortex, the SH ozone median (184 nmol mol[-1]) was only 54% of that in the NH (341 nmol mol[-1]), while in low depletion years, SH ozone median (214 nmol mol[-1]) reached 57% of the NH values (371 nmol mol[-1]). Notably, the SSW events (2002 and 2019) increased SH UTLS ozone by 24% (43 nmol mol[-1]) compared to high depletion years, while in the NH, the increase was 9% (31 nmol mol[-1]).

## 1 Introduction

Ozone (O$_3$) has generated an effective radiative forcing (ERF) of +0.47 [0.24 to 0.71] W m$^{-2}$ over the industrial era, from 1750 to 2019. The ERF for ozone represents 17% of the total anthropogenic ERF change, estimated at +2.72 [1.96 to 3.48] W m$^{-2}$ (Forster et al., 2021). The ozone abundance in the troposphere is related to the atmosphere's oxidizing capacity and is closely associated with methane abundance, the second strongest greenhouse gas, due to hydroxyl radical (·OH) reactions (Saunois et al., 2020). At ground level, short-term exposure to ozone impairs human lung function and contributes to developing asthma symptoms in susceptible people (McConnell et al., 2002; Zheng et al., 2021). Total global mortality due to long-term ozone exposure has been estimated at 365,000 [175,000 to 564,000] deaths in 2019 (Health Effects Institute, 2020).

The tropospheric ozone budget is controlled by chemical production and loss, dry deposition and stratosphere-troposphere exchange (STE) (Archibald et al., 2020). Oxidation of volatile organic compounds (VOC) initiated by hydroxyl radicals lead to tropospheric ozone formation. During this process, hydroperoxyl (HO$_2$) and alkyl peroxy radicals (RO$_2$) reactions with nitric oxide (NO) yield nitrogen dioxide (NO$_2$), which photolyzes to produce ozone (Monks et al., 2015). In the troposphere, the chemical loss results mainly from ozone photolysis ($\lambda \leq 320$ nm) and ozone reaction with hydroxyl and hydroperoxyl radicals (Atkinson, 2000). An important ozone sink is dry deposition at the surface (Clifton et al., 2020).

The STE, estimated at 284 ±193 Tg O$_3$ year$^{-1}$ (Griffiths et al., 2021; Szopa et al., 2021), is comparable with the net chemical ozone production in the troposphere. STE involves several mechanisms, including tropopause folds within mid-latitude cyclones, cutoff-low decay and gravity wave breaking (Archibald et al., 2020; Cooper et al., 2004; Stohl et al., 2003). STE shows strong geographical asymmetry and seasonal cycles. STE dominates in the extratropics, with hotspots in mountain regions of North and South America, over the Tibetan Plateau and in the storm tracks over the North Atlantic Ocean and North Pacific Ocean, which are more intense in winter and early spring in both hemispheres (Škerlak et al., 2014).

The poleward expansion of the SH Hadley circulation has also been proposed as a mechanism able to enhance downward transport of ozone from the stratosphere to the troposphere at higher latitudes (Lu et al., 2019). STE's contribution to the tropospheric ozone budget is expected to increase due to declining levels of ozone-depleting substances (ODS) and the Brewer-Dobson circulation (BDC) acceleration, especially in the lower stratosphere, caused by increasing greenhouse gas emissions (Butchart, 2014). In contrast, simulations based on climate models do not show BDC acceleration in the mid-stratosphere of the mid-latitude NH since the late 1980s due to declining ODS and the timing of volcanic eruptions (Garfinkel et al., 2017).

Ozone changes in the upper troposphere-lower stratosphere (UTLS) strongly impact the radiative forcing of the atmosphere (Riese et al., 2012; Skeie et al., 2020). Therefore, in situ measurements in the UTLS, characterized by bidirectional trace gas exchange (Gettelman et al., 2011), are crucial to reducing the uncertainty of chemistry climate models and validating satellite retrievals (Bourgeois et al., 2020). Unfortunately, in the Southern Hemisphere, measurements from IAGOS (In-service Aircraft for a Global Observing System: Petzold et al., 2015) commercial aircraft rarely extend beyond 35ºS, and only a few aircraft-based research missions have characterized UTLS chemical composition in this area. The HALO (High Altitude and LOng

range) aircraft measured dry air masses descending from the stratosphere down to an altitude of 7 km a.s.l. over the Antarctic in September 2012 (Rolf et al., 2015). Also, flights over the South Atlantic and South Pacific Oceans conducted by the Atmospheric Tomography (ATom) mission measured a higher frequency of stratospheric intrusions during spring (Bourgeois et al., 2020). In addition, valuable long-term vertical ozone profiles have been obtained through ozonesonde launches at remote sites in higher latitudes of the SH, such as the Ushuaia (Argentina), Lauder (New Zealand) and Macquarie Island (Australia) stations, since 2008, 1986 and 1994 respectively (Zeng et al., 2024; Van Malderen et al., 2024).

In this context, the research project Southern Hemisphere Transport, Dynamics, and Chemistry (SouthTRAC) was designed to comprehensively study the dynamical and chemical processes in the SH UTLS region using the HALO instrumented aircraft during the late winter and spring seasons (September-November) of 2019 (Rapp et al., 2021). SouthTRAC coincided with a polar stratospheric sudden warming (SSW) event that occurred in September 2019 (Rao et al., 2020; Rapp et al., 2021). SSW in the SH is a phenomenon that is unusual compared with the higher frequency of SSW events in the NH, with approximately one event per decade in the SH versus six in the NH (Dunn et al., 2020; Charlton and Polvani, 2007). SSW is caused by wave events that can decelerate the westerly polar night jet and warm the polar stratosphere by descending motion, resulting in reduced heterogeneous active chlorine formation onto polar stratospheric clouds and less catalytic ozone depletion (Scambos and Stammerjohn, 2020). Accordingly, the annual mean total ozone reported in 2019 was 65 DU higher than the long-term average, at 60º-90ºS (Dunn et al., 2020). The mission also coincided with intense Australian bushfires and biomass-burning events in central South America, which produced plumes detected by HALO in the UTLS (Kloss et al., 2021; Johansson et al., 2022).

Given the multiple dynamical and chemical processes that influence ozone in the UTLS (Millán et al., 2024; Bourgeois et al., 2020; Neu et al., 2014; Riese et al., 2012) and the interhemispheric differences in processes such as the magnitude of the stratospheric ozone depletion and frequency of SSW events, this study, investigates hemispheric differences in UTLS ozone in air masses of stratospheric origin, with focus on mid-latitudes, where STE plays a dominant role in determining ozone levels. We also leverage the increased ozone abundance under low-depletion conditions derived from SSW events to determine the intrahemispheric UTLS ozone differences. Through this comparative analysis, we aim to quantify the influence of two processes, i.e., stratospheric ozone depletion and SSW, on the ozone UTLS, where this species is an important radiative forcer. Our analysis utilized stratospheric and tropospheric chemical tracers measured during the SouthTRAC mission, by IAGOS commercial aircraft and by ozonesondes. Spatial coverage was further enhanced using the Copernicus Atmosphere Monitoring Service reanalysis (CAMSRA), which we compared against in situ measurements. Furthermore, by characterizing ozone and chemical species exchange in the UTLS, we identified a valuable dataset for evaluating the models that simulate ozone transport from the stratosphere to the troposphere in both hemispheres.

## 2 Data

The UTLS ozone measurements used in this study are available from (research and commercial) aircraft and ozonesondes. These in situ measurements were complemented with vertical ozone profiles obtained from chemical reanalysis. To determine the stratospheric or tropospheric origin of the ozone data, we used water vapor (or humidity) measurements from the aircraft and radiosondes coupled to the ozonesondes and from the chemical reanalysis in addition to carbon monoxide (CO), nitric acid ($HNO_3$) and hydrogen chloride (HCl) measurements from some of the aircraft data. In the next sections, we provide more details on these datasets.

### 2.1 SouthTRAC data

The SouthTRAC mission was conducted mainly over the southernmost region of South America within the latitudinal and meridional band 45º-60ºS and 30º-85W, respectively (**Fig. 1**). The mission had two stages: 9 Sep-6 Oct 2019 and 6-15 Nov 2019, totaling 16 flights, coinciding with the maximum ozone depletion in Antarctica. During this mission, the HALO aircraft was equipped with 13 instruments for sampling the physical and chemical properties of the atmosphere. This research focuses on the measurements of ozone, carbon monoxide, water vapor, nitric acid and hydrogen chloride. The number of measurements of each species is presented in **Table 1**.

In situ ozone was measured by the Fast AIRborne Ozone monitor (FAIRO) provided by the Karlsruhe Institute of Technology (KIT). FAIRO combines two measurement principles: UV ozone absorption at 255 nm and dry chemiluminescence from the reaction of ozone with an organic dye (coumarin) at 500 nm (Ermel et al., 2013). In this dual configuration, the UV photometer calibrated the chemiluminescence response. We utilized ozone records measured by chemiluminescence due to their higher sampling frequency (5 Hz) compared with UV photometry (0.25 Hz). Typical one standard deviation ($1\sigma$) measurement precision is 0.08 nmol mol$^{-1}$ for the UV spectrometer measuring at 0.25 Hz and ~0.05 nmol mol$^{-1}$ for the chemiluminescence detector measuring at 5 Hz, 1 bar and 5 nmol mol$^{-1}$ absolute mixing ratio (Zahn et al., 2012). The estimated uncertainty for UV photometer measurements was 2% and for the combined techniques (UV and chemiluminescence) was 2.5%.

The University of Mainz Airborne Quantum Cascade Laser Spectrometer (UMAQS) measured in situ carbon monoxide. This instrument is based on direct absorption in the near-infrared through a continuous-wave quantum cascade laser (Müller et al., 2015). The instrument had a typical carbon monoxide precision of 0.68 nmol mol$^{-1}$ ($2\sigma$), an accuracy of 1.2 nmol mol$^{-1}$ and a total uncertainty of 1.4 nmol mol$^{-1}$.

Static pressure and static temperature were measured by Basic HALO Measurement and Sensor System (BAHAMAS) with an estimated accuracy of 0.3 hPa and 0.5 K, respectively (Giez et al., 2017; Kaufmann et al., 2018). BAHAMAS also measured water vapor at the absorption line of 1.37 μm using the Sophisticated Hygrometer for Atmospheric ResearCh (SHARC), a tunable diode laser hygrometer. The limit of detection of SHARC measurement achieved during SouthTRAC campaign was 2 μmol mol$^{-1}$ with an absolute uncertainty of ±1 μmol mol$^{-1}$ (Kaufmann et al., 2018).

Nitric acid and hydrogen chloride were measured using the Airborne Chemical Ionization Mass Spectrometer (AIMS). In this system, the ion source generates reagent ions ($SF_5^-$) that selectively react with $HNO_3$ and $HCl$ to form $HFNO_3^-$ and $HFCl^-$, respectively. A fast quadrupole mass spectrometer then separates the ion products based on their mass-to-charge ratio. Typical accuracy, precision ($1\sigma$), and limit of detection are 16%, 20%, and 20 pmol mol$^{-1}$ for $HNO_3$ and 12%, 16%, and 15 pmol mol$^{-1}$ for $HCl$, respectively (Jurkat et al., 2016).

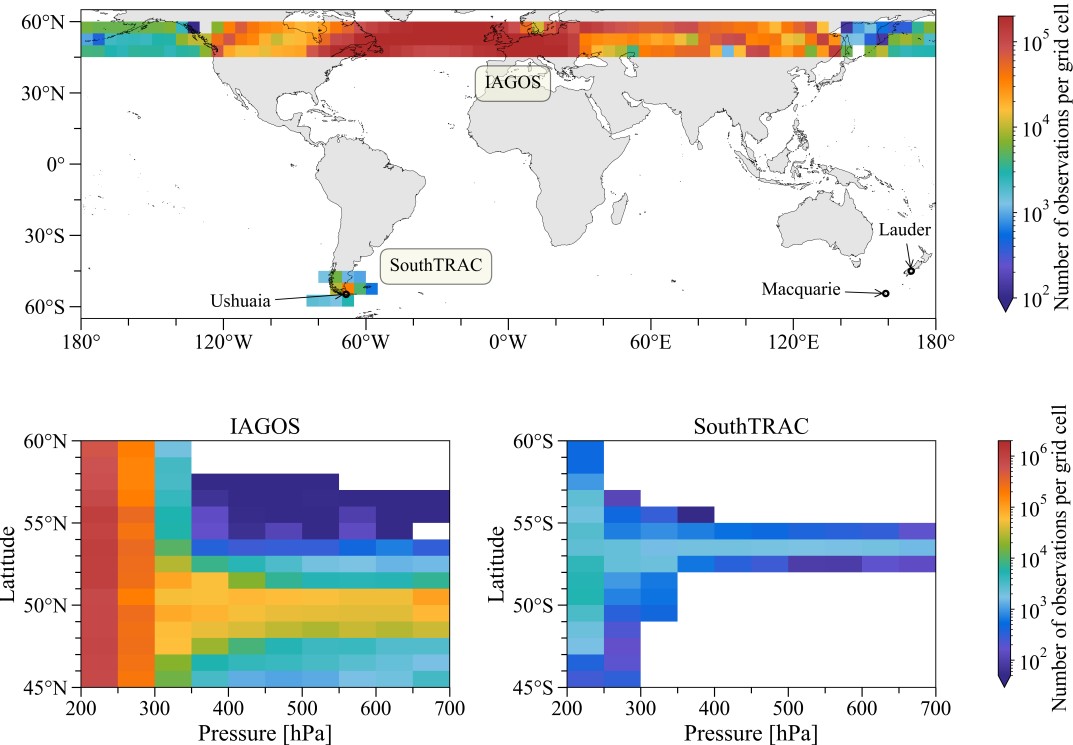

**Figure 1: Routes position frequency obtained from aircraft GPSs. All panels show only SouthTRAC and IAGOS data within the same latitudinal band 45°-60° in the SH and NH. SouthTRAC shows data from 4 Sep to 20 Nov 2019 and IAGOS between 4 Mar and 20 May from 2002 to 2022. The upper panel shows the number of GPS observations by grid cell (5° x 5°) between 700 and 200 hPa. The lower panels show the number of GPS observations in every grid cell (1° x 50 hPa) between 700 and 200 hPa for IAGOS (left) and SouthTRAC (right). The upper panel also indicates the location of the ozonesonde sites of Ushuaia, Lauder and Macquarie Island.**

## 2.2 Ozonesonde data

Ozone profiles from ozonesondes launched at Ushuaia, Argentina (54.85 °S, 68.31 °W) and Macquarie Island, Australia (54.50 °S, 158.95°E) were obtained from the World Ozone and Ultraviolet Radiation Data Centre, WOUDC (https://woudc.org/) while ozone profiles from Lauder, New Zealand (45.04 °S, 169.68 °E) were obtained from the Tropospheric Ozone Assessment Report – Phase II (TOAR-II) Focus Working Group HEGIFTOM (Harmonization and Evaluation of Ground-Based Instruments for Free Tropospheric Ozone Measurements). These data were included in our research in order to compare SouthTRAC measurements to historical data.

These ozonesondes were coupled with radiosondes, which measured pressure, temperature, RH, wind speed and wind direction. **Table 1** shows the available periods utilized in this study and the number of validated flights from each site. The ozonesondes were linearly interpolated to an altitude grid with intervals of 50 m (Ohyama et al., 2018).

During 2002-2022, several radiosonde models were utilized interchangeably (e.g., Lauder). The Vaisala RS92 and its replacement RS41 sensor are among the most frequently used models. Regarding RH, the latter performs better than the older version, particularly in cloudy conditions (Jensen et al., 2016). Overall, both sensors have an uncertainty of 5% and show a consistent difference of about 1-2%, with the RH of the RS41 almost always greater than RS92 (Jensen et al., 2016).

**Table 1**: Summary of data used. The columns detail dataset sources, spatial coverage, temporal periods, the number of flights in the period and the number of measurements for each variable (in ozonesondes is the number of interpolated measurements) between 200-300 hPa.

| Data source | Hemisphere or site (lat., lon.) | Periods | No. flights or valid launches | Variable (No. measurements) |
|---|---|---|---|---|
| SouthTRAC | SH (45ºS-60ºS, 30ºW-85ºW) | 4 Sep - 20 Nov 2019 | 16 | Pressure: 43k<br>$O_3$: 39k<br>CO: 26k<br>$H_2O$: 43k<br>RH: 43k |
| WOUDC | 1. Ushuaia (54.85ºS, 68.31ºW),<br>2. Lauder (45.04ºS, 169.68ºE),<br>3. Macquarie (54.50ºS, 158.95ºE) | 1. 2008-2022<br>2. 2002-2022<br>3. 2002-2022 | 1. 141<br>2. 232<br>3. 203 | 1. Pressure: 7.1k<br>$O_3$: 7.1k<br>RH: 7.1k<br>2. Pressure: 12k<br>$O_3$: 12k<br>RH: 12k<br>3. Pressure: 11K<br>$O_3$: 11k<br>RH: 11k |
| IAGOS | NH (45°N-60°N) | 2002-2022 | 6,315 | Pressure: 16M<br>$O_3$: 11M<br>CO: 11M<br>$H_2O$: 14M<br>RH: 15M |

## 2.3 IAGOS data

The IAGOS research infrastructure provides in situ measurements of chemical species on board several commercial aircraft. IAGOS is a continuation of the former observation systems MOZAIC (Measurement of Ozone and Water Vapor by Airbus In-Service Aircraft: Marenco et al., 1998) and also includes the CARIBIC measurement system (Civil Aircraft for the Regular Investigation of the Atmosphere Based on an Instrument Container: Brenninkmeijer et al., 1999, 2007; Stratmann et al., 2016), now called IAGOS-CORE and IAGOS-CARIBIC.

The MOZAIC observations started in August 1994 for ozone and water vapor on board five equipped aircraft, whereas the carbon monoxide measurements started in December 2001. The CARIBIC observations started in 1997 for a wide range of chemical species, including ozone, water vapor and carbon monoxide, on a single aircraft.

IAGOS-CORE provides ozone (and carbon monoxide) data using an ultraviolet (infrared) absorption spectrometer, with an accuracy, precision and time response of 2 nmol mol$^{-1}$, 2% and 4 s (5 nmol mol$^{-1}$, 5%, 30 s) respectively (Nédélec et al., 2003; Nédélec et al., 2015; Thouret et al., 1998). Water vapor is measured with a capacitive hygrometer, characterized by an accuracy of 5% in RH with respect to liquid water (RHL), or 6% RHL in the vicinity of the thermal tropopause at midlatitudes, and a
time response of 5 – 300 s (Helten et al., 1998; Neis et al., 2015; Rolf et al., 2024; Smit et al., 2014).

IAGOS-CARIBIC uses an identical instrument to measure ozone as described for HALO (see section 2.2). Carbon monoxide data are measured by a commercial resonance fluorescence UV instrument modified for use onboard commercial aircraft. Its accuracy, precision and time response are respectively less than 2 nmol mol$^{-1}$, 1 – 2 nmol mol$^{-1}$, and less than 2 s (Scharffe et al., 2012). Water vapor measurements are performed with a photoacoustic laser spectrometer and a frost-point hygrometer,
with an accuracy of less than 4 % or 0.3 μmol mol$^{-1}$ for the mixing ratio. The time response varies between 5 and 30 s for humid (>100 ppm) and dry air masses (<10 ppm) respectively (Dyroff et al., 2015; Zahn et al., 2014).

## 2.4 CAMS & ERA5 reanalysis

We utilized vertical ozone profiles, temperature and specific humidity from the chemical reanalysis CAMSRA (Copernicus Atmosphere Monitoring Service, reanalysis). CAMSRA has 60 model levels from the surface to the top of the atmosphere, a
horizontal resolution of 0.75° x 0.75° and a time resolution every 3 hours (Inness et al., 2019).  We also retrieved zonal wind at 60°N (60°S) and 10 hPa from ERA5 reanalysis (Hersbach et al., 2023). ERA5 has hourly resolution, 0.25° x 0.25° horizontal resolution and 37 pressure levels. We used post-processed daily mean on pressure levels obtained from Copernicus Climate Change Service (C3S) Climate Data Store (CDS).

## 3 Method

In this section, we describe how we used the data available to analyze late winter mid-spring UTLS ozone differences at the mid-latitudes between both hemispheres. We first describe our study period and the used UTLS definition (section 3.1), then we mention how we distinguish between high and low stratospheric ozone depletion years (section 3.2) and show how we ascertain the stratospheric origin of the analyzed UTLS ozone mixing ratios (section 3.3).

## 3.1 Study period & UTLS definition

In this study, we denominate the UTLS as the segment between 200 (~12 km) and 300 (~9 km) hPa, while the free troposphere is the segment between 700 to 300 hPa. We focus on the period between late winter and mid-spring in both hemispheres: 4 Sep–20 Nov (SH) and 4 Mar–20 May (NH). For comparison, IAGOS data from 2002 to 2022 were analyzed for the same latitudinal band and altitude range in the NH. **Figure 1** shows the sampling density given by the aircraft's positional data to identify those regions of the atmosphere with more data availability. In addition, **Figure S1** in the Supplement depicts the

highest number of GPS observations obtained between 300 and 200 hPa compared to the range of 700 to 300 hPa.

## 3.2 High and low ozone depletion years definition

We distinguished low ozone depletion years from high ozone depletion years according to the occurrence of SSW events. Therefore, we applied the definition proposed by Charlton and Polvani (2007) and used ERA5 data to detect major SSW, which is based on determining the reversal of the daily-mean, zonal-mean zonal winds from westerly to easterly at 60ºN and

10 hPa from November to April. **Table 2** shows the so-called central date (Charlton and Polvani, 2007), i.e., the first day on which the daily zonal mean zonal wind at 10 hPa and 60°N changed from westerly to easterly between November and March. During the detection procedure, we required 20 consecutive days with westerly winds before identifying another event. We excluded cases with easterly zonal winds that did not return to westerly for at least 10 consecutive days before 30 April. In the SH, the shift from westerly to easterly is considered between July and October, and we excluded the cases when the wind did

not return to westerly by 30 November.

In addition to the 2002 major SSW identified by the above metric, this research also considered the 2019 SH SSW as a low ozone depletion year. While the 2019 event did not reverse the zonal wind at 60ºS and 10 hPa, the weakening of the stratospheric polar vortex was as strong as that of the 2002 event (Lim et al., 2021; Rao et al., 2020).

Additionally, given that the ozone-depleted air masses typically reach their minimum ozone content in October (March) of the

Southern (Northern) hemisphere, we split the flights into late winter-early spring and mid-spring, representing the latter, the period when depleted ozone air likely reaches the SouthTRAC study region (**Table 2**).

**Table 2**: Study period considered for both hemispheres, subperiods and low ozone depletion years according to SSW definition.

| Study period | Late winter-early spring (no. flights or launches) | Mid spring (no. flights or launches) | Low-depletion years (central date) |
|---|---|---|---|
| 4 Mar - 20 May | 4 - 31 Mar (2142 IAGOS flights) | 1 Apr – 20 May (4173 IAGOS flights) | 2002 (17 Feb), 2003 (18 Jan), 2004 (5 Jan), 2006 (21 Jan), 2007 (24 Feb), 2008 (22 Feb), 2009 (24 Jan), 2010 (24 Mar), 2013 (6 Jan), 2018 (12 Feb), 2019 (1 Jan) and 2021 (5 Jan) |
| 4 Sep - 20 Nov | 4 - 30 Sep (10 SouthTRAC flights) (204 launches) | 1 Oct - 20 Nov (6 SouthTRAC flights) (372 launches) | 2002 (25 Sep) and 2019[†] (15 Sep) |

[†]: Central date determined when the zonal-mean zonal winds at 60°S and 10 hPa decrease to $\leq 20$ m/s (Rao et al., 2020).

## 3.3 Stratospheric character determination

Stratospheric and tropospheric tracer scatterplots provide a magnitude of bidirectional exchange across the tropopause by the identification of stratospheric and tropospheric branches and mixed air regions according to their tracer abundances (Gettelman et al., 2011). Hence, in this study, our first analysis used tracer correlations to compare the northern (IAGOS) and southern hemisphere (SouthTRAC) mid-latitudes, and between high and low ozone depletion years (NH).

We utilized several sensors with different accuracies to filter air masses by RH. Vaisala radiosonde sensors have a longer response time, and their vertical resolution highly depends on temperature; thus, radiosonde sensors are less accurate than SHARC, IAGOS-CORE, and IAGOS-CARIBIC measurements. Therefore, we tested different relative humidity levels, from 50% to 10% (every 10 intervals), to effectively filter tropospheric moist mid-latitude air masses. We estimated that a RH lower than 20% is effective in excluding tropospheric air masses richer in water vapor and is sufficiently conservative, given the sensor accuracy. This threshold behaves similarly to the 50 nmol mol$^{-1}$ carbon monoxide filter (analyzed in more detail in Section 4.2).

CAMSRA profiles were also filtered using 20% RH for the 45°N-60°N and 45°S-60°S zonal bands between 2003 and 2022 (Table 1). In this case, we calculated the relative humidity with respect to liquid water from temperature and specific humidity.

## 4 Results and discussion

In the following section, we used tracer correlations to characterize the tropospheric or stratospheric origin of the air masses in the UTLS and free troposphere of both hemispheres. In section 4.2, we applied filters mainly based on the relative humidity to assign the stratospheric origin in the UTLS and then determine the inter- and intrahemispheric ozone differences. Finally, in section 4.3, we compare the results based on situ ozone measurements with CAMSRA outputs.

### 4.1 Tropospheric and stratospheric tracer correlations

Tracer-to-tracer scatterplots between stratospheric ozone tracer and tropospheric tracers, such as water vapor and carbon monoxide at two pressure intervals, are shown in **Figures 2** and **3**. In general, we observed higher ozone values in the UTLS and free troposphere of the NH compared to the SH. Scatterplots show that ozone between 300-200 hPa in the SH rarely exceeds 400 nmol mol$^{-1}$. In contrast, in the NH, high ozone levels near 1200 nmol mol$^{-1}$ associated with low water vapor levels (i.e., drier air) are observed. The same pattern is observed between 700 and 300 hPa, with ozone levels in the NH much higher than observed in the SH (up to an excess of 200 nmol mol$^{-1}$). **Figure 2** also shows a similar relationship between ozone and water vapor in low and high-depletion years at both pressure intervals in the NH. In the following section, we clearly distinguish the ozone differences between the low and high-depletion years.

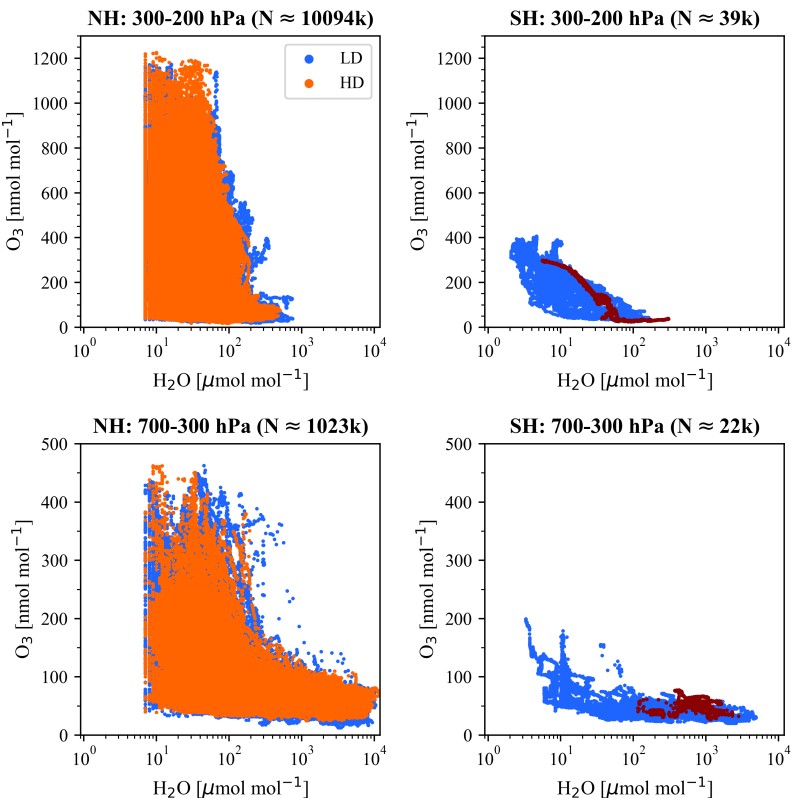

**Figure 2: Scatter plots of ozone versus water vapor. The left panels show scatter plots for the period between 4 Mar and 20 May from 2002 to 2022 for the NH between 300-200 and 700-300 hPa (IAGOS data set described in Table 1). The right panels show the same for the SH between 4 Sep and 20 Nov 2019 (SouthTRAC data described in Table 1). Low and high-depletion (LD and HD) years are in blue and orange colors, respectively. Above each plot, N stands for the number of observations. In the SH, the flight conducted on 12 Nov 2019 is highlighted in dark red.**

Tracer-to-tracer scatterplots of ozone and carbon monoxide show that in the NH, carbon monoxide reaches higher levels (~300 nmol mol[-1]) at both 300-200 and 700-300 hPa (**Figure 3**). Notice that some carbon monoxide values are not included in NH because they were not simultaneously measured with ozone. Again, carbon monoxide and ozone show a similar relationship in low and high-depletion years in the NH. In contrast, in the SH, carbon monoxide rarely exceeds 100 nmol mol[-1] in both pressure intervals (300-200 hPa and 700-300 hPa). An exception to this behavior was observed during the flight conducted on 12 Nov 2019 (mid-spring), which detected elevated levels of carbon monoxide (>200 nmol mol[-1]), ozone (>100 nmol mol[-1]) and water vapor (>100 μmol mol[-1]) at pressures below 300 hPa. These elevated levels were attributed to tropospheric plumes and may have been pyrogenic in origin, given the simultaneous bushfires in Australia that extended from Sep 2019 to Mar 2020 (Kloss et al., 2021). Ultimately, the impact of emissions from the Australian bushfires in the UTLS may be interpreted as feedback initiated early in August-September 2019, which derived from the deceleration of the polar vortex in the middle stratosphere.

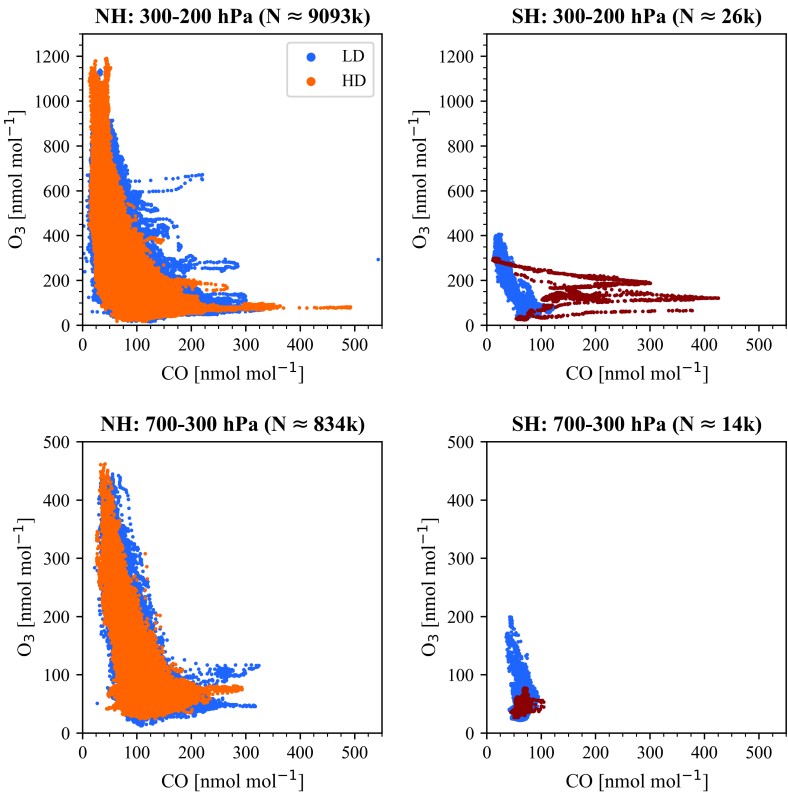

Figure 3: Scatter plots of ozone versus carbon monoxide. The left panels show scatter plots for the period between 4 Mar and 20 May from 2002 to 2022 for the NH between 300-200 and 700-300 hPa (IAGOS data set described in Table 1). The right panels show the same for the SH between 4 Sep and 20 Nov 2019 (SouthTRAC campaign data described in Table 1). Low and high-depletion (LD and HD) years are in blue and orange colors, respectively. Above each plot, N stands for the number of observations. In the SH, the flight conducted on 12 Nov 2019 is highlighted in dark red.

In this regard, the abundance of nitric acid and hydrogen chloride are also useful in determining the plume's origin since these two species are stable in the stratosphere but more prone to removal in the troposphere. Tracer-to-tracer scatterplots of these species versus ozone and carbon monoxide (**Fig. 4**) show that nitric acid and hydrogen chloride were generally higher when ozone was higher and lower when carbon monoxide was higher. This anticorrelation between carbon monoxide and stratospheric tracers suggests that this plume predominantly originates from the troposphere. Nevertheless, we cannot entirely exclude the mixing processes of stratospheric and tropospheric air masses, as shown by the fluctuations registered during the flight (**Fig. 4**).

Overall, in this section, we found that the NH does not show differences between high and low depletion years. Therefore, in the following section, we test different filters to isolate the air of stratospheric origin in the UTLS and thus investigate its ozone mixing ratios during high and low depletion years in both hemispheres. In this section we also showed that the NH has higher carbon monoxide levels than the SH and a threshold of 50 nmol mol[-1] at 300-200 hPa seems to well represent the stratospheric branch in both Hemispheres (**Fig. 3**). Accordingly, we utilized the carbon monoxide levels (<50 nmol mol[-1]) in the following

section to distinguish stratospheric air masses, even though our primary classification relies on the RH threshold.

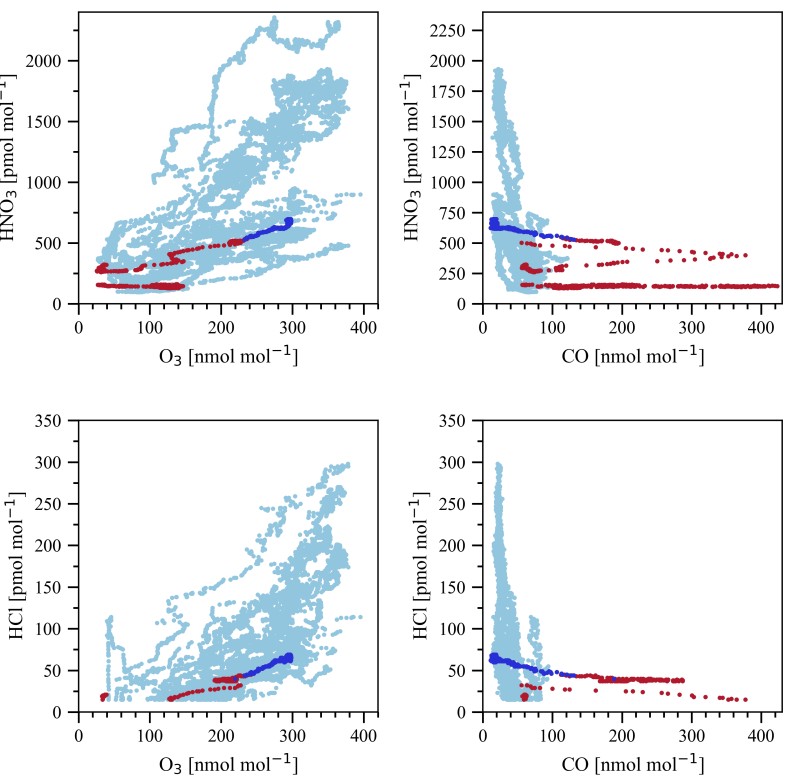

Figure 4: Scatter plots of nitric acid versus ozone, nitric acid versus carbon monoxide, hydrogen chloride versus ozone and hydrogen chloride versus carbon monoxide. These scatter plots are based exclusively on SouthTRAC data. Measurements obtained during the flight conducted on 12 Nov 2019 (ST25) are highlighted from the other flight measurements depending on their relative humidity in blue (< 20%) or in red (> 20%).

## 4.2 Inter and intrahemispheric ozone differences in the UTLS

As discussed earlier in the methodology, aircraft RH observations are more accurate than radiosondes. Therefore, we tested several RH filters on the air masses, aiming to minimize the impact of the reported error of the sensors within the UTLS and to minimize the inclusion of tropospheric air. The following discussion is based on a 20% RH filter, while results for 30% and 10% RH are shown in the Supplement **(Fig. S2 and S3)**. **Figure 5** depicts ozone boxplots and the corresponding observation numbers (in the 10-base logarithm) by pressure bins. It is worth noting that the pressure bins between 300 and 280 hPa have less data, particularly those obtained from aircraft sampling **(Fig. 5)**, which implies a lower representativity at the lowest altitude bins. The same observation applies to the 230-220 hPa bin during mid-spring in the SouthTRAC mission. In the appendix **(Fig. A1)**, we also present the ozone boxplots without filters to visualize the aspect of the original data. The filter's applicability can be clearly verified for the subperiod late winter-early spring in the SH, where differences between high and low ozone depletion years can be noticed without any filter (Figure A1). However, we can also note that the 20% RH filter is

very effective at removing the lowest ozone mixing ratios with tropospheric origin (or mixed air).

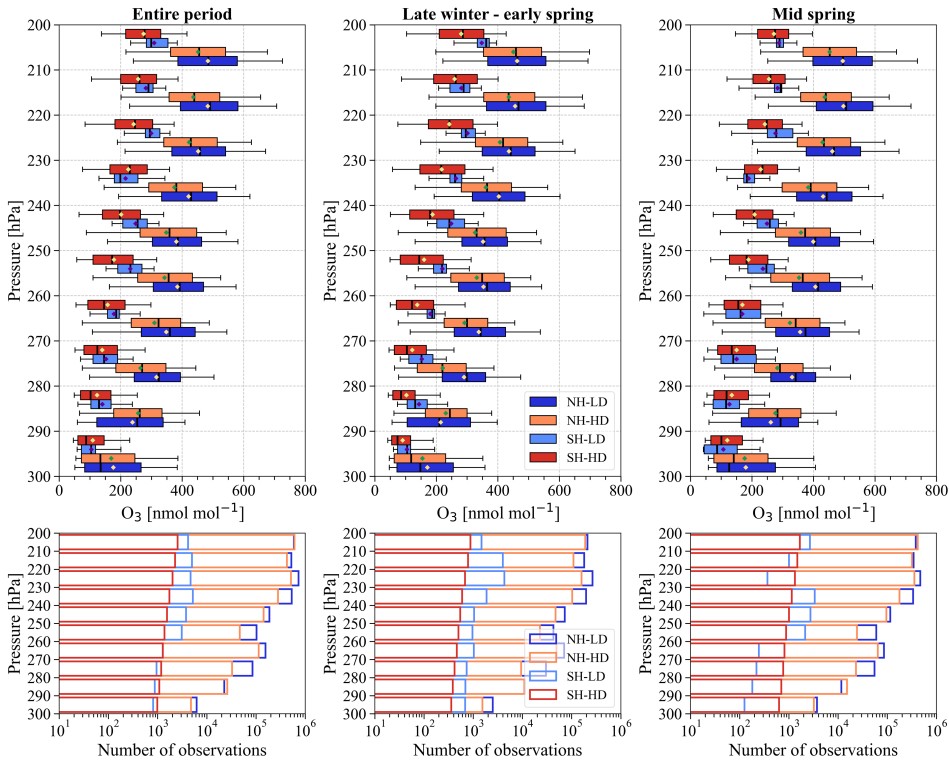


**Figure 5: Ozone box plot (nmol mol⁻¹) and associated sample sizes by pressure bins. The plots include ozone data with RH less than 20% obtained from IAGOS, SouthTRAC and WOUDC (Table 1). In the upper panel, the left, middle and right plots show the entire period, late winter-early spring and mid-spring respectively, for the northern and southern hemispheres and high-depleted (HD) and low-depleted (LD) years. Box plots indicate the median, the 25th and 75th percentiles. The diamonds in the upper panels indicate**
**the mean values. The whiskers extend to the 5th and 95th percentiles. In the lower panels, the X-axis shows the observation numbers. The rest of the plot information is the same as described in the upper panel. High and low-depletion years in SH included ozonesondes launched from Ushuaia, Lauder and Macquarie Island.**

We observed higher ozone abundances in the NH for the entire period analyzed and for both subperiods, i.e., late winter-early spring and mid-spring compared with the SH (**Fig. 5**). We also noticed differences between high and low-depletion years in
both hemispheres. To provide a quantitative magnitude of these differences, we averaged the medians between 300-200 hPa (shown in Figure 5) and then calculated the ozone difference (and standard deviation) between low and high depletion years. In the NH, ozone in low depletion years increased by 9% (31 ± 11 nmol mol⁻¹) compared with high depletion years. The changes in the subperiods late winter-early spring and mid-spring were similar, with 8% (26 ± 22 nmol mol⁻¹) and 10% (36 ± 12 nmol mol⁻¹), respectively.

In the SH, ozone the difference between high and low-depletion years was more intense in late winter-early spring, especially for pressures less than 270 hPa. Within 300-200 hPa, the difference between medians averaged 53 ± 17 nmol mol⁻¹, being on average 31% greater relative to high depletion years (**Fig.5**). The observed change in mid-spring was more variable. We

attribute this behavior to the shorter time of flight in some pressure bins. On the other hand, 2019 is heavily influenced by the SouthTRAC sample size despite including three ozonesonde sites, as shown in **Figure B1**, where the ozone boxplots are

without SouthTRAC data. Interestingly, by excluding SouthTRAC data, the mid-spring period is consistently higher in low depletion years at all pressure bins and reproduced ozone's higher difference in late winter-early spring. Hence, considering only ozonesondes, the enhancement for mid-spring was 17% ($33 \pm 10$ nmol mol$^{-1}$) and for the entire period (SouthTRAC combined with ozonesondes) it was 24% ($43 \pm 13$ nmol mol$^{-1}$).

The higher ozone levels observed in the late winter-early spring during the 2002 and 2019 SH SSW (45º-60ºS), compared with

high depletion years (2002-2022), clearly illustrate the impact of the early warming of the polar vortex in September 2019, which was even earlier than the September 2002 warming (Scambos and Stammerjohn, 2020). In this region, Antarctic ozone depletion, more intense between 12 and 20 km, typically begins in August, accelerates in September, and reaches a minimum in October when ozone starts to recover. Therefore, the period of higher ozone destruction rate in September was entirely captured by the SouthTRAC mission.

Although the magnitude of the SSW event was not categorized as a major event, i.e., easterly winds did not reverse at 10 hPa at 60º S, it was strong enough to decrease the halogen-catalyzed ozone loss over the polar region and be detected in subpolar regions as shown in **Figure 5**. Later, in mid-spring, the observed ozone changes were less intense. Nonetheless, it is worth noting that even during this less ozone-depleted atypical scenario in 2019, the ozone abundance in the UTLS was not comparable to the higher levels observed in the NH, representing only 57% of the NH ozone median.

The agreement between carbon monoxide and 20% RH filters varies with pressure bins, with a better agreement at pressures less than 270 hPa and where more aircraft measurements were available. On average, between 270 and 200 hPa, the percentage ozone difference between both filters was $13 \pm 16\%$ in low depletion years (NH), $15 \pm 8\%$ in high depletion years (NH) and $5 \pm 6\%$ in low depletion years (SH). In the SH, we have compared only the low-depletion years because carbon monoxide was measured only by the SouthTRAC mission. Thereby, **Figure C1** shows the ozone abundances by pressure bins after applying

the carbon monoxide filter (<50 nmol mol-1). Similarly to the 20% RH filter, less ozone abundances were observed in the SH during low-depletion years compared with the NH, and a similar pattern can be observed for the ozone difference between high and low-depletion years in the NH.

**4.3 Outputs comparison based on in situ measurement and reanalysis**

So far, we have provided ozone mixing ratios (in drier air) for the entire longitudinal band without paying attention to spatial

asymmetries previously reported in the literature (Škerlak et al., 2014). Therefore, the motivation of this section is to compare the in situ measurements of ozone, filtered by 20% RH in the UTLS, with the values produced by the CAMS reanalysis over the entire longitudinal band within 45-60° latitude and for low and high depletion years indicated in **Table 2**, aiming to provide a spatially resolved perspective.

In **Figure 6**, the ozone boxplot is displayed in five pressure bins, determined by the five vertical levels available for CAMSRA in 300-200 hPa. **Figure 6** shows that the reanalysis reproduces the ozone interhemispheric differences, i.e., higher ozone in the NH and differences between high and low ozone depletion years. We also compared the pattern agreement between the ozone boxplots (CAMSRA) and in situ measurements. **Figure 6** clearly illustrates the similarity between the ozone medians obtained from the CAMSRA and in situ measurements (SouthTRAC and ozonesondes) particularly in the period with the highest number of flights, i.e., late winter-early spring, for pressures lower than 270 hPa. The lower agreement between the ozone medians of CAMSRA and in situ measurements in mid-spring period was determined by the lower flight frequency in this period, i.e., six flights during the mid-spring period versus ten in the late winter-early spring period (also discussed in section 4.2). Regardless, we can observe the similarity of the ozone medians obtained from ozonesondes and the reanalysis for the entire period (less affected by the mid-September SouthTRAC data).

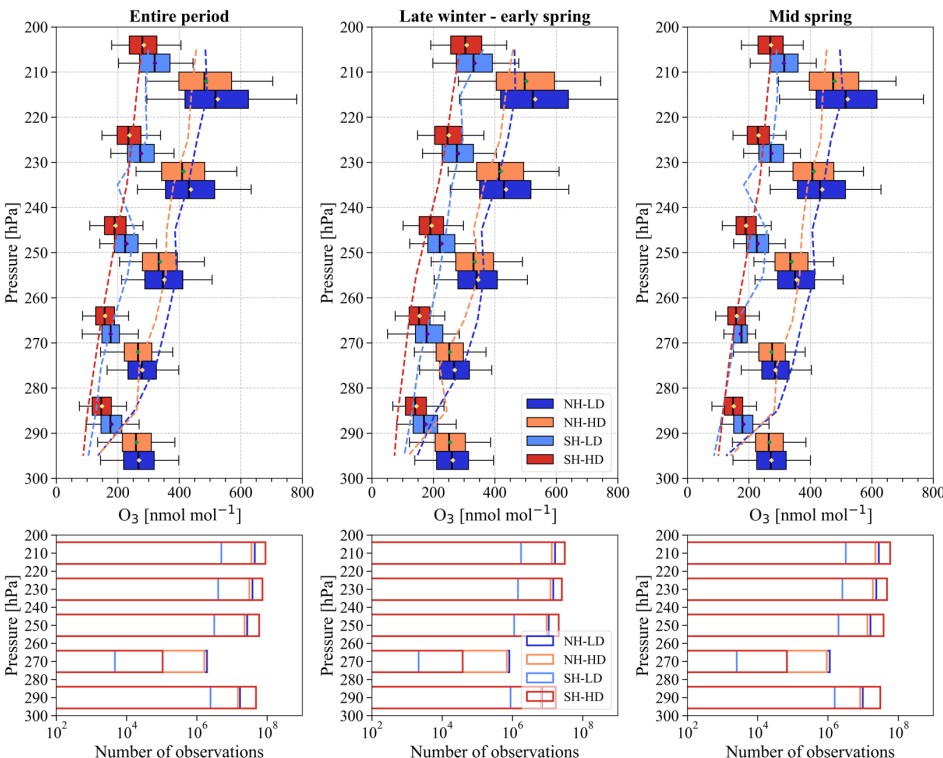

**Figure 6: Ozone box plot (nmol mol$^{-1}$) and associated sample sizes by pressure bins obtained from the CAMS reanalysis. The plots include ozone data with RH less than 20%. The vertical dashed lines are the ozone medians obtained through in situ measurements previously shown in Figure 5. In the upper panel, the left, middle and right plots show the entire period, late winter-early spring and mid-spring, respectively, for the northern and southern hemispheres and high-depleted and low-depleted years. Box plots indicate the median, the 25th and 75th percentiles. The diamonds in the upper panels indicate the mean values. The whiskers extend to the 5$^{th}$ and 95$^{th}$ percentiles. In the lower panels, the X-axis shows the observation numbers. The rest of the plot information is the same as described in the upper panel.**

**Figure 7** shows CAMS reanalysis across the entire NH and SH longitudinal band for high and low depletion years from 2003 to 2022. Consistently, the spatial distribution of ozone median (filtered by 20% RH) in the UTLS obtained by CAMSRA

reproduced the hemispheric differences discussed previously. Mean UTLS ozone during high and low ozone depletion years

was 399 and 425 nmol mol$^{-1}$ in the NH, and 228 and 265 nmol mol$^{-1}$ in the SH. In addition, **Figure 7** shows some relevant features related to ozone exchange in the UTLS. In the NH, we observe two regions where the ozone was relatively higher due to lower stratospheric ozone depletion: eastern North America (60°W) and Notheast Asia (140°E). In the SH, under low stratospheric ozone depletion, we observe a large region with relatively higher ozone levels over the southern Indian Ocean and, to a minor degree, the South Pacific Ocean.

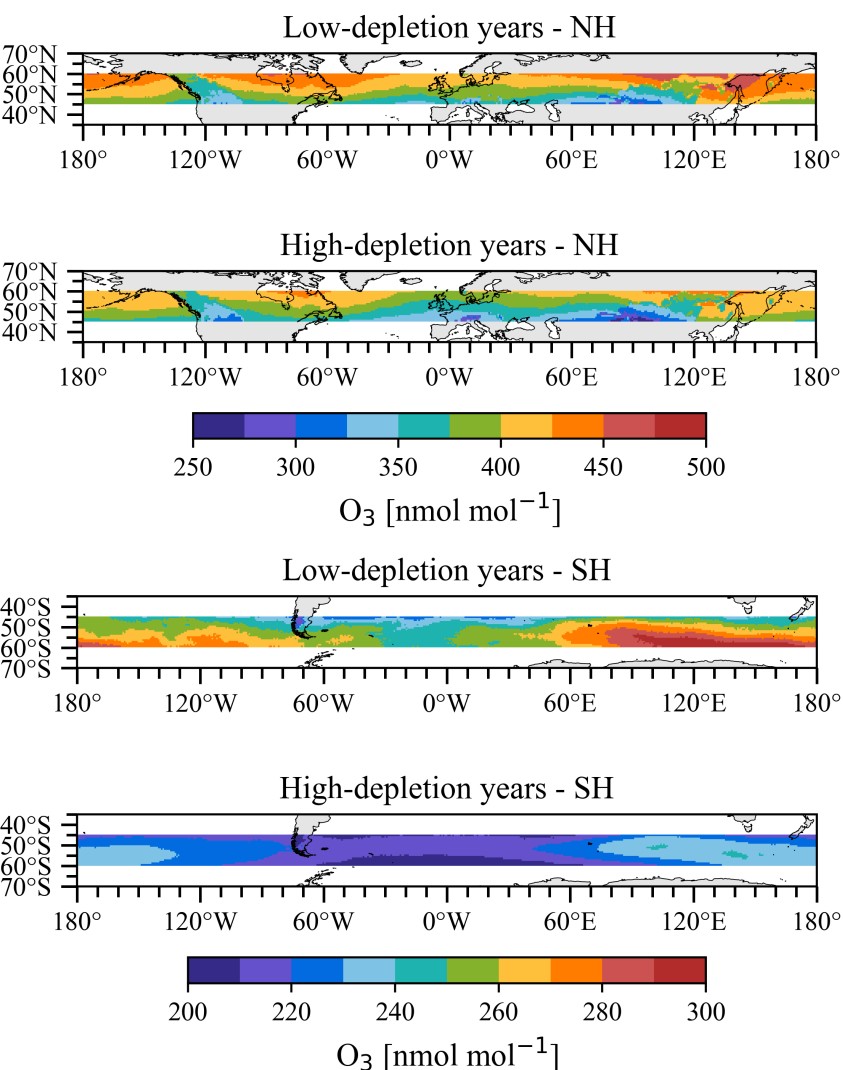

Figure 7: Ozone medians (nmol mol$^{-1}$) obtained from CAMS reanalysis filtered by RH less than 20% at a resolution of 0.75º x 0.75º and between 300-200 hPa. The upper panel depicts the ozone median in UTLS of the northern hemisphere, separated into low and high depletion years (entire period), while the lower panel describes the same for the southern hemisphere. Note that the color scales are different in the NH and SH given the strong differences in ozone mixing ratios.


**Conclusions**

The SouthTRAC mission represents an invaluable contribution to atmospheric chemistry research in the Southern Hemisphere. This aircraft campaign provided unique data recorded during a rare stratospheric sudden warming event in the Southern Hemisphere that produced profound effects on the chemical structure of the atmosphere, leveraged in this research as a proxy for a diminished ozone depletion scenario. The main conclusions of our research are listed below.

Tracer-to-tracer scatterplots between stratospheric ozone and tropospheric tracers (carbon monoxide and water vapor) are
consistent with previous airborne campaign studies, which show high ozone and low carbon monoxide levels in the Southern Hemisphere (SH) in both the free troposphere and the upper troposphere-lower stratosphere (UTLS), which in this case was influenced by distant wildfires in Australia.

The bidirectional exchange observed in the UTLS during the 12 Nov 2019 flight provides evidence that transport from tropospheric sources can also influence this region of the atmosphere. During this flight, low levels of nitric acid (<500 pmol
mol$^{-1}$) and hydrogen chloride (<50 pmol mol$^{-1}$) were consistently measured, alongside higher levels of carbon monoxide (>200 nmol mol$^{-1}$) and elevated water vapor (>100 μmol mol$^{-1}$), indicating a strong tropospheric signature in the UTLS.

The comparative analysis of ozone mixing ratios, with stratospheric origin between 300-200 hPa and 45-60° latitude, shows:

i. Lower ozone levels in the SH UTLS compared with the Northern Hemisphere (NH). In high ozone depletion years, defined by the strength of the stratospheric vortex, the ozone median in SH UTLS was 54% of that in the NH UTLS. During years
impacted by early stratospheric sudden warming, i.e., low ozone depletion years, the SH UTLS ozone median was 57% of the NH UTLS values.

ii. The difference between high and low ozone depletion years in the NH UTLS (March-May) as a function of SSW events resulted in a 9% ozone enhancement (31 ± 11 nmol mol$^{-1}$) compared with years without SSW. This enhancement was larger in the SH, reaching 24% (43 ± 13 nmol mol$^{-1}$).

iii. The chemical reanalysis is especially useful in areas with sparse in situ observations, such as the SH, and its application in this research allows us to assess the representativeness of in-situ observations. Moreover, CAMSRA reproduced the ozone interhemispheric differences and the difference between high and low ozone depletion years in each hemisphere. Additionally, the CAMSRA-based analysis identified the southern Indian Ocean as the region with higher ozone in the UTLS.

Finally, differences in spatial coverage between the in situ measurements in the two hemispheres may affect the
representativeness of the results and should be considered when interpreting these comparisons. Overall, this study emphasizes the need to expand highly resolved ozone profiles in the upper troposphere and lower stratosphere, particularly across the Southern Hemisphere, where these measurements are crucial for accurately quantifying current stratosphere-troposphere exchange processes and their impact on the tropospheric ozone budget.

## Appendix A: Ozone boxplot without RH filter

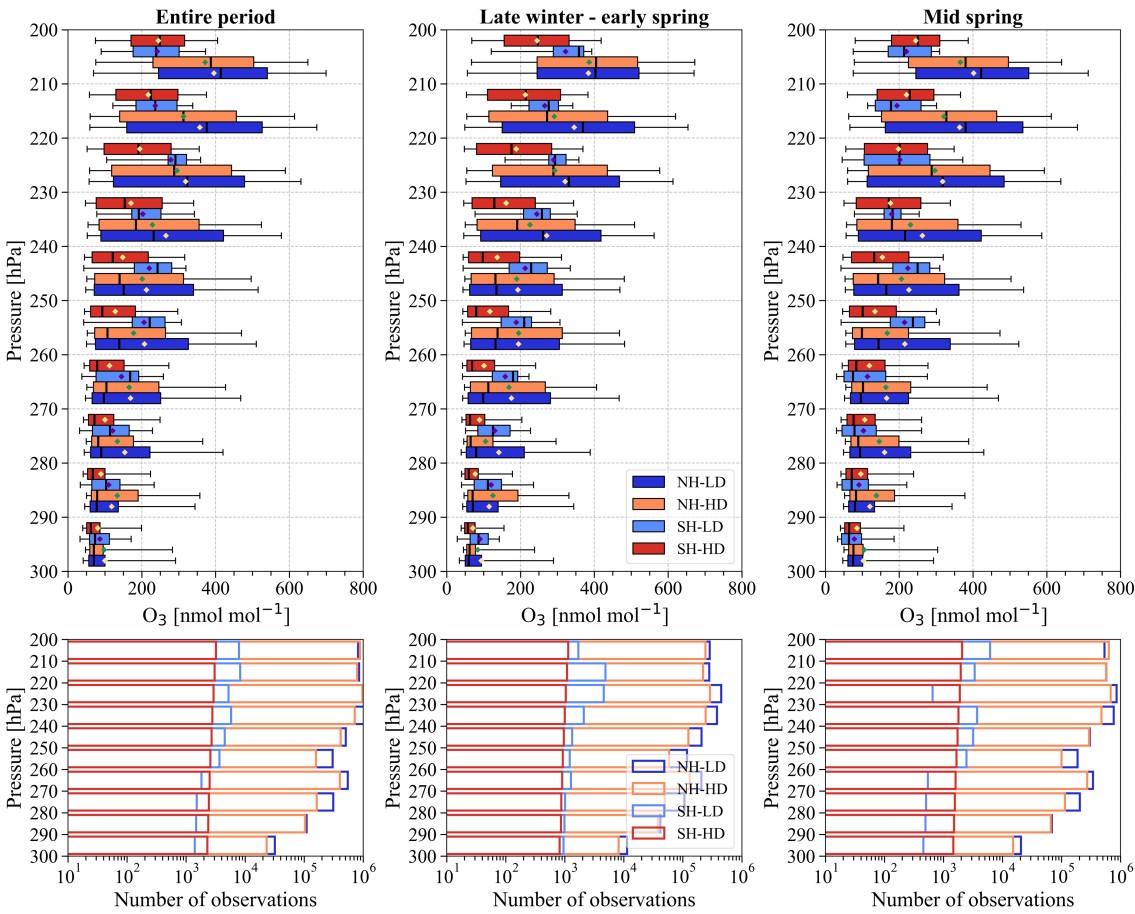

**Figure A1: Ozone box plot (nmol mol[-1]) and associated sample sizes by pressure bins. The plots include ozone data without RH less than 20% obtained from IAGOS, SouthTRAC and WOUDC (Table 1). In the upper panel, the left, middle and right plots show the entire period, late winter-early spring and mid-spring respectively, for the northern and southern hemispheres and high-depleted (HD) and low-depleted (LD) years. Box plots indicate the median, the 25th and 75th percentiles. The diamonds in the upper panels indicate the mean values. The whiskers extend to the 5th and 95th percentiles. In the lower panels, the X-axis shows the observation numbers. The rest of the plot information is the same as described in the upper panel. High and low-depletion years in SH included ozonesondes launched from Ushuaia, Lauder and Macquarie Island.**

## Appendix B: Ozone boxplot without SouthTRAC data

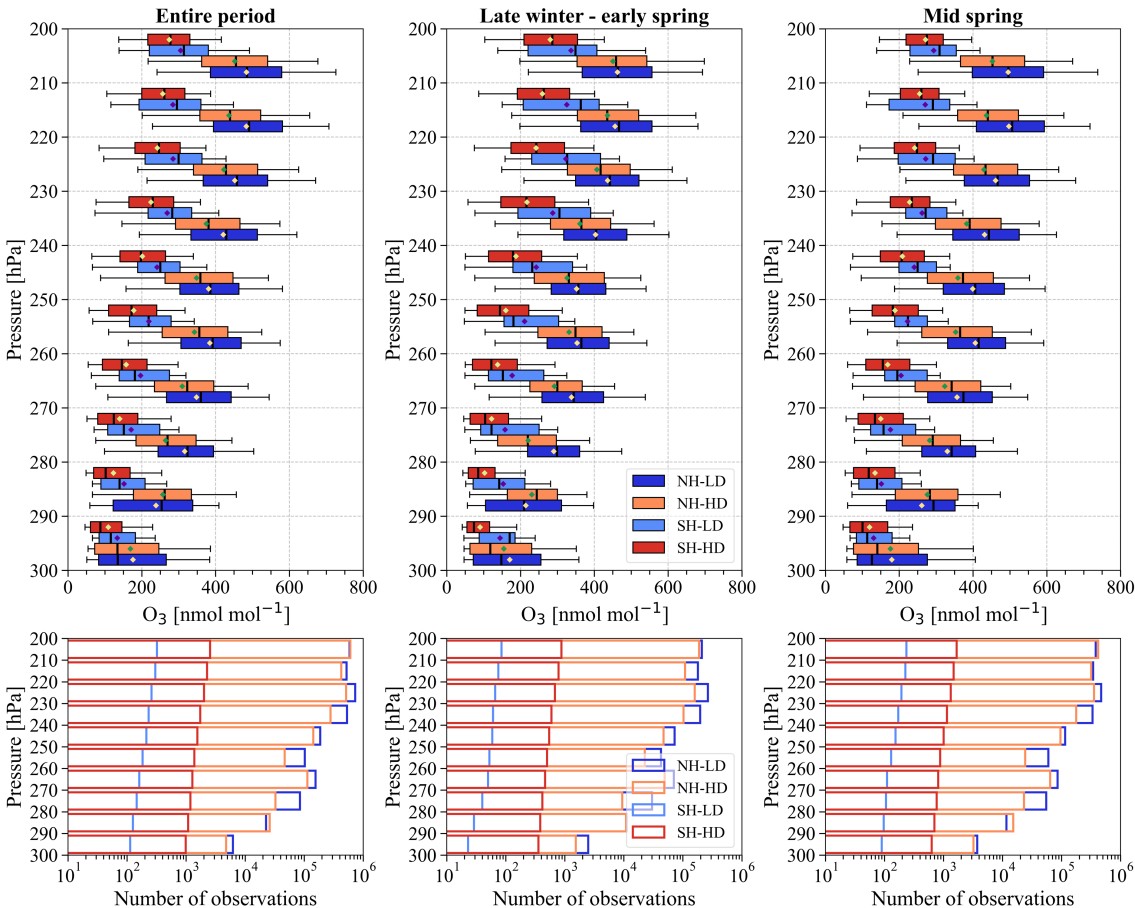

**Figure B1: Ozone box plot (nmol mol[-1]) and associated sample sizes by pressure bins. The plots include ozone data with RH less than 20% obtained exclusively from IAGOS and WOUDC (Table 1). In the upper panel, the left, middle and right plots show the entire period, late winter-early spring and mid-spring respectively, for the northern and southern hemispheres and high-depleted (HD) and low-depleted (LD) years. Box plots indicate the median, the 25th and 75th percentiles. The diamonds in the upper panels indicate the mean values. The whiskers extend to the 5th and 95th percentiles. In the lower panels, the X-axis shows the observation numbers. The rest of the plot information is the same as described in the upper panel. High and low-depletion years in SH included ozonesondes launched from Ushuaia, Lauder and Macquarie Island.**

425

430

435

## Appendix C: Air masses filtered by carbon monoxide

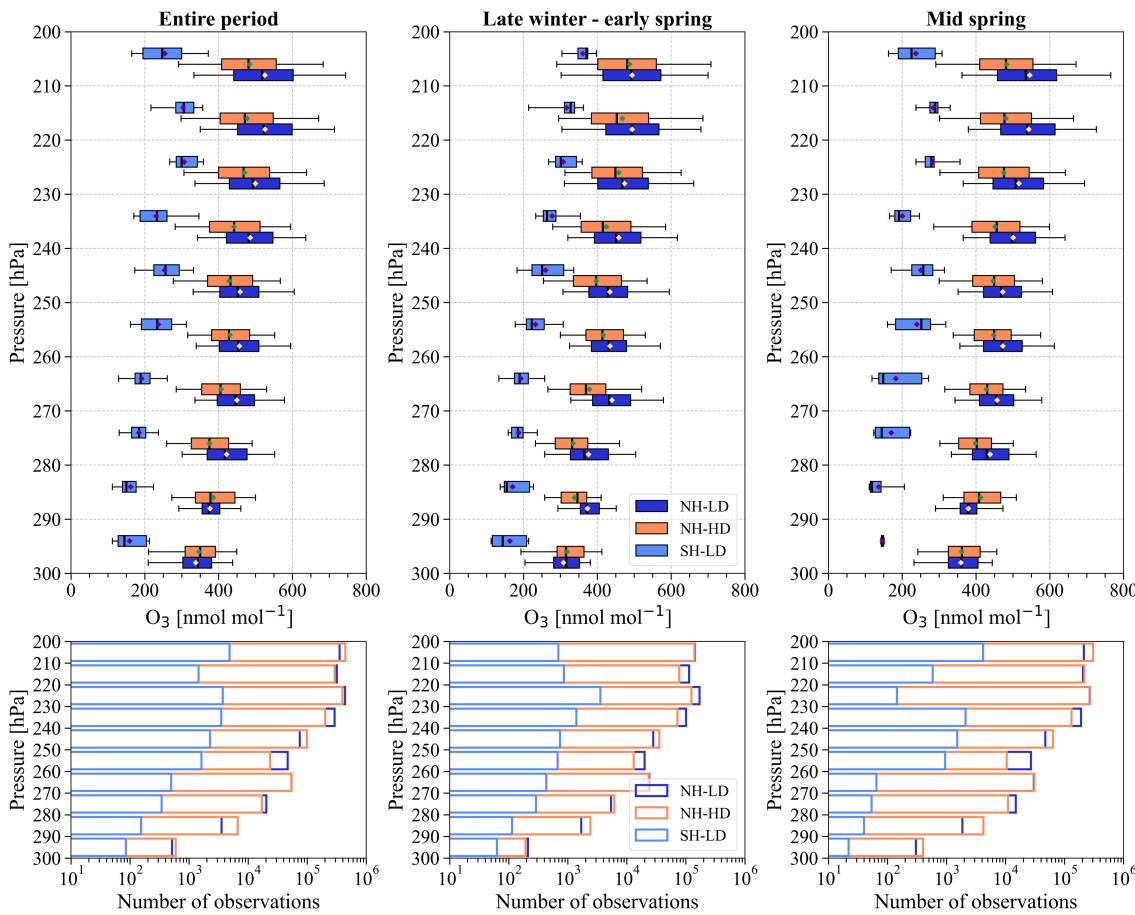

**Figure C1: Ozone box plot (nmol mol$^{-1}$) and associated observation numbers by pressure bins. The plots include ozone data with CO less than 50 nmol mol$^{-1}$. In the upper panel, the left, middle and right plots show the entire period, late winter-early spring and mid-spring for the northern and southern hemispheres and high-depleted and low-depleted years. Box plots indicate the median, the 25th and 75th percentiles. The diamonds in the upper panels indicate the mean values. The whiskers extend to the 5th and 95th percentiles. In the lower panels, the X-axis shows the observation numbers. The rest of the plot information is the same as described in the upper panel.**

*Data availability*. SouthTRAC data can be accessed via the HALO database under https://halo-db.pa.op.dlr.de. IAGOS-CARIBIC data can be retrieved at https://zenodo.org/doi/10.5281/zenodo.8188548 (IAGOS-CORE / MOZAIC can be quoted via IAGOS-DB) (Zahn et al., 2024). Ozonesonde data for Ushuaia and Macquarie station can be retrieved from the World Ozone and Ultraviolet Radiation Data Centre (WOUDC) at http://woudc.org/archive/Archive-NewFormat/OzoneSonde_1.0_1/stn339/ecc (WOUDC, 2024). Ozonesonde records for Lauder station were obtained from: Harmonization and Evaluation of Ground Based Instruments for Free Tropospheric Ozone Measurements (HEGIFTOM) at https://hegiftom.meteo.be (HEGIFTOM, 2025). CAMSRA data were obtained from Atmospheric Data Store (ADS): https://ads.atmosphere.copernicus.eu/datasets/cams-global-reanalysis-eac4?tab=overview (ADS, 2025). ERA5 reanalysis were obtained from Copernicus Climate Change Service (C3S) Climate Data Store (CDS) at https://cds.climate.copernicus.eu/datasets/derived-era5-pressure-levels-daily-statistics (C3S/CDS, 2025).

*Author contribution*. RS, OC, YC, & CO: conceptualization. OC, CO, YC & RS: methodology. CO & YC: data curation. RS, CO, YC, & OC: formal analysis. RS: writing - original draft preparation. All authors: writing – review & editing. BS & PH initiated and coordinated the SouthTRAC campaign with the HALO research aircraft.

*Competing interest*. The contact author has declared that none of the authors has any competing interests.

*Special issue statement*. This article is part of the special issue "Tropospheric Ozone Assessment Report Phase II (TOAR-II) Community Special Issue (ACP/AMT/BG/GMD inter-journal SI)". It is a result of the Tropospheric Ozone Assessment Report, Phase II (TOAR-II, 2020–2024).

*Acknowledgments*. The authors acknowledge the collaboration provided by the Center for Climate and Resilience Research for the analysis contained in this article (ANID/FONDAP/1523A0002), the supercomputing infrastructure of the NLHPC (ECM-02). The authors thank the pilots, engineers, and scientists from DLR Flight Experiments for their excellent support during the SouthTRAC mission. The authors thank the SouthTRAC member Martin Zoeger (BAHAMAS data). The authors acknowledge that MOZAIC/CARIBIC/IAGOS data were created with support from the European Commission, national agencies in Germany (BMBF), France (MESR), and the UK (NERC), and the IAGOS member institutions (http://www.iagos.org/partners). The participating airlines (Lufthansa, Air France, Austrian, China Airlines, Hawaiian Airlines, Air Canada, Iberia, Eurowings Discover, Cathay Pacific, Air Namibia, Sabena) supported IAGOS by carrying the measurement equipment free of charge since 1994. The data are available at http://www.iagos.fr thanks to additional support from AERIS. The authors thank the following PIs for providing ozonesonde and radiosonde data: Roeland Van Malderen (HEGIFTOM), Richard Querel (Lauder station), Gerardo Carbajal (Ushuaia station) and Matt Tully (Macquarie Island station).

*Financial support*. This research has been partially supported by ANID (National Agency for Research and Development of Chile, grant no. ANID/FONDAP/1523A0002; ANID-DFG/FONDEQUIP-SouthTRAC/DFG190003; ANID/FONDECYT/1241459. JGU acknowledges funding by the German Science Foundation (Deutsche Forschungsgemeinschaft, DFG) Priority Program SPP 1294: HO4225/15-1, and HO4225/14-1

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
