# Peer review of "Hemispheric differences in ozone across the stratosphere-troposphere exchange region"

_EGUsphere, 2024_

## Referee Comment (RC2)

Review of "Hemispheric differences in ozone across the stratosphere-troposphere exchange region" by Rodrigo J. Seguel, et al.

**Summary and General Comments:**

This paper examines 2019 SouthTRAC aircraft and 2008-2018 Ushuaia ozonesonde data in the midlatitude Southern Hemisphere (SH), and IAGOS aircraft data in the Northern Hemisphere (NH) in various years to understand hemispheric differences in stratosphere-to-troposphere exchange, sudden stratospheric warming (SSW) events, and lower stratospheric ozone. The authors use the heavily instrumented SouthTRAC HALO aircraft to determine criteria for separating air of tropospheric and stratospheric origin in the UTLS. Ultimately, carbon monoxide (50 nmol mol-1) and relative humidity (20%) are used to make this distinction and extend the analysis of STE events beyond just 2019.

The authors identify years with (low depletion year) and without (high depletion year) SSW events, separate the ozone profile data from aircraft and ozonesondes into the low and high depletion years, and examine the differences in both hemispheres during STE events. Note that I am not sure I saw explicitly how the SSWs were defined (major SSW? A slowing of the 60N/S 10hPa winds to some magnitude?). They find that air of stratospheric origin in the SH UTLS contains much lower ozone than the NH for both the low and high depletion years. The analysis of the aircraft and ozonesonde data is supported by the CAMS chemical reanalysis showing this result throughout the entire 45-60N/S latitude band.

The text is polished and the figures are easy to digest and well-constructed. The SouthTRAC September-November 2019 aircraft data are serendipitous in that you can examine a rare Antarctic SSW and low depletion event with multiple in-situ chemical species (including some aged fire plumes). However, this paper is missing a lot of the broader context that would be provided by analysis of the Lauder (45S) and Macquarie Island (55S) ozonesonde records. Both of those records predate 2000 and would enable you to also examine the 2002 Antarctic SSW event, effectively doubling your SH low-depletion cases. Those two stations are on nearly the complete opposite side of the globe, adding longitudinal contrasts that are important, as you show in Figure 6. In fact, you might end up with different results with the inclusion of more SH data and a second SSW year in 2002. The SouthTRAC and Ushuaia ozone data are close to a local minimum in ozone according to Figures 1 and 6. In either case, you will add confidence to your results.

The study periods for the NH and SH should be aligned, something like 2002-2022 for both hemispheres, in my opinion. There should be plenty of IAGOS data in the NH to extend the study, and of course there is a dense network of ozonesonde stations in Europe and North America in your 45-60N band of interest. At the very least, why not align the study periods for both hemispheres and include the 2019 Ushuaia data?

Finally, I am curious to know who oversees the Ushuaia station and if they were offered co-authorship. Often, this recognition can help maintain a station's support.

**Recommendation:**

In my assessment, this paper is not ready for publication and needs major revisions. The addition and analysis of more datasets, alignment of the study periods in both hemispheres, and a broader context and discussion of the impact of the differing STE ozone amounts between the two hemispheres will make this an important contribution.

**Line-by-Line and Technical Comments:**

Abstract lines 15-17: The answer to the question raised in the first sentence of the Abstract is well known and does not accurately capture what I think you are trying to demonstrate in this manuscript: Even during low depletion SH years, Southern Hemisphere UTLS ozone in STE events is far less than Northern Hemisphere high depletion years. The NH vs. SH lower stratospheric ozone differences in these latitude bands are obvious from the global ozonesonde network. See my quick analysis in Figure R1:

[Figure]

**Figure R1: Average ozone partial pressure profiles from select ozonesonde sites in the 45-60° latitude band in both Northern and Southern Hemispheres.**

Abstract Line 26: "...the SSW event increased SH UTLS ozone by 37%..."

Lines 82-83: Again, we can refer to Figure R1 to demonstrate that SH STE events will contain less ozone than NH STE events. There is simply less ozone in the SH mid-latitudes.

Line 97: It is contained in Table 1, but please state in the main text what time periods you are focusing on. Note also my concerns about mismatched time periods for the NH and SH.

Line 104: "...interannual variability of both hemispheres..."

Table 1: Why are the time periods examined not the same for NH and SH? Surely there are enough NH IAGOS data to make the analysis period for both hemispheres 2008-2022 or so? Also, extending back to 2002 will enable you to add a second SH SSW event. By adding Lauder and Macquarie Island ozonesonde data to the analysis, this can easily be accomplished for the Southern Hemisphere as well.

Line 142 and other locations: RS92 radiosondes

Lines 167 and 168: The RH measurements come from the Vaisala radiosondes, not the ozonesonde instrument

Figure 2: It looks to me like the water vapor measurements are rounded to the nearest whole µmol mol-1, but are there really measurements of just 1 µmol mol-1?

Line 219: "tracers" not "traces"

Line 226: Again, the RH measurements are from the radiosondes attached to the ozonesondes

Line 232: "...applies to the 230-220 hPa bin..."

Figure 4: Why did you not include 2019 Ushuaia data in addition to SouthTRAC?

Line 242: Are these differences statistically significant? It might be difficult to draw conclusions based on the fairly limited data from single SH low depletion case in 2019.

Figure 6: Is the UTLS here also defined as 300-200 hPa? Please state clearly in the Figure 6 caption or plots

Figure 6 Caption: "...the same for the Southern Hemisphere"

Line 287-288: This sentence is unclear. Please rewrite.

Lines 321-323: There is certainly a lack of in-situ ozone profile data in the SH, but you are not taking full advantage of what is available in this latitude band, including the Lauder and Macquarie Island ozonesonde datasets

---

## Author Response (AR1)

April 8, 2025

**Author Comments (egusphere-2024-3719)**

Manuscript title: Hemispheric differences in ozone across the stratosphere-troposphere exchange region

> We have carefully read the referee and community comments. We greatly appreciate their quality and constructiveness. Accordingly, we have addressed each comment and incorporated the suggested changes in a new version of the manuscript. The referee and community comment revisions are addressed below.

**Referee Comment (RC1)**

**General comments:**

The paper is well written, the data analysis is thorough and detailed. However, I sometimes miss the more global background a bit. Therefore, a justification why it is important to study hemispheric differences in UTLS ozone, and possible causes and consequences should be further elaborated on.

The major shortcoming of the study is the very limited spatial extent of the study region in the southern hemisphere, due to the availability of measurements, so that the representativeness of the hemispheric differences (and their longitudinal dependence) is hard to assess. In this context, I found it very surprising that the authors did not consider to include also the Lauder (45°S) ozonesonde dataset in their analysis, next to the Ushuaia ozonesonde dataset, which is just at the border of the defined zonal band (45°-60°S). The authors did mention the Lauder ozonesonde dataset at Line 69. Some explanation for not including this dataset is missing here.

Also, the time period for the available measurements used for the NH (IAGOS: 2011, 2018-2022) does not fully align with the time periods used for the SH (2019: SouthTRAC, 2008-2018: WOUDC). Why not using the entire IAGOS dataset from 2008-2022 for the NH and also using the WOUDC data for Ushuaia 2019-2022? Wouldn't there be an impact of a possible temporal mismatch between the NH and SH UTLS ozone observations that are compared? This could be investigated with the CAMSRA, if you would take full advantage of its entire temporal extent (see also in the specific comments). And why did you not compare the 2019 UTLS ozone observations between SouthTRAC and Ushuaia?

> **Answer**: Although we will address the above-mentioned aspects in the specific comments, at this point, we indicate that we extended the analysis period and included two additional ozonesondes launched within the latitudinal band of interest. In summary:

> - IAGOS: 2002-2022
> - Ozonsonde launched from Lauder and Macquarie Island: 2002-2022
> - CAMSRA: 2003-2022

I think section 2 should be better organized as well and a distinction should be made between 2. Data (section 2, now subsections 2.2, 2.3, 2.4, 2.5) and 3. Methodology (now subsections 2.1 and 2.5). Currently, you are presenting and mentioning the geographical (horizontal and vertical) extent of the observations, before presenting the observations themselves. To me, this is not a very logical order.

**Answer**: Thanks for suggesting this improvement. Now, we have two sections, one for data and the other for the methodology:

2. Data
2.1 SouthTRAC data
2.2 Ozonesonde data
2.3 IAGOS data
2.4 CAMS & ERA5 reanalysis
3. Method
3.1 Study period & UTLS definition
3.2 High and low ozone depletion years definition
3.3 Stratospheric character determination

At several locations, some additional clarifications are needed, which are summed up in the specific comments.

**Specific comments:**

Lines 80-85: I assume much more literature is available and much more key findings regarding the hemispheric UTLS ozone differences. Please add those here.

**Answer**: We added more literature related to SSWs, e.g.:

▪ *Charlton, A. J. and Polvani, L. M.: A new look at stratospheric sudden warmings. Part I: Climatology and modeling benchmarks, J Clim, 20, https://doi.org/10.1175/JCLI3996.1, 2007.*
▪ *Lim, E. P., Hendon, H. H., Butler, A. H., Thompson, D. W. J., Lawrence, Z. D., Scaife, A. A., Shepherd, T. G., Polichtchouk, I., Nakamura, H., Kobayashi, C., Comer, R., Coy, L., Dowdy, A., Garreaud, R. D., Newman, P. A., and Wang, G.: The 2019 southern hemisphere stratospheric polar vortex weakening and its impacts, https://doi.org/10.1175/BAMS-D-20-0112.1, 2021.*
▪ *Rao, J., Garfinkel, C. I., White, I. P., and Schwartz, C.: The Southern Hemisphere Minor Sudden Stratospheric Warming in September 2019 and its Predictions in S2S Models, Journal of Geophysical Research: Atmospheres, 125, https://doi.org/10.1029/2020JD032723, 2020.*

And also key papers addressing the importance of ozone changes in UTLS, e.g:

▪ *Millán, L. F., Hoor, P., Hegglin, M. I., Manney, G. L., Boenisch, H., Jeffery, P., Kunkel, D., Petropavlovskikh, I., Ye, H., Leblanc, T., and Walker, K.: Exploring ozone variability in the upper troposphere and lower stratosphere using dynamical coordinates, Atmos. Chem. Phys., 24, 7927–7959, https://doi.org/10.5194/acp-24-7927-2024, 2024.*
▪ *Neu, J. L., Hegglin, M. I., Tegtmeier, S., Bourassa, A., Degenstein, D., Froidevaux, L., Fuller, R., Funke, B., Gille, J., Jones, A., Rozanov, A., Toohey, M., Von Clarmann, T., Walker, K. A., and Worden, J. R.: The SPARC data initiative: Comparison of upper troposphere/lower stratosphere ozone climatologies from limb-viewing instruments and the nadir-viewing tropospheric emission spectrometer, J Geophys Res, 119, https://doi.org/10.1002/2013JD020822, 2014.*

On the other hand, the time period extension (from 2002) and more long-term ozonesondes (159º and 170ºE) added to the analysis allow us to improve the context to articulate a broader discussion. Also, at the beginning of the abstract and the end of the introduction, we added some lines to describe the motivation of the research:

Abstract: *"Ozone changes in the upper troposphere-lower stratosphere (UTLS) resulting from dynamical and chemical processes strongly affect the atmosphere's radiative forcing. This study analyzed intra- and interhemispheric ozone differences in the UTLS within the 45-60° latitude band, distinguishing between years disrupted by sudden stratospheric warming (SSW) events from 2002 to 2022…"*

Introduction: *"Ozone in the UTLS is highly variable, driven by a complex interplay of dynamical and chemical processes (Millán et al., 2024; Bourgeois et al., 2020; Neu et al., 2014; Riese et al., 2012). In this context, SSW events represent a diminished ozone depletion scenario, which, combined with the highly resolved profiles obtained during the SouthTRAC campaign, provides a unique and realistic framework for assessing ozone changes. In this study, we leverage the increased ozone abundance under low-depletion conditions derived from SSW events to determine the effect on the ozone mixing ratio in UTLS of both hemispheres. This comparative analysis is based on stratospheric and tropospheric chemical traces measured during the SouthTRAC mission, by IAGOS commercial aircraft and by from ozonesondes, focusing on the 45-60° latitude band. Spatial coverage is further enhanced using the Copernicus Atmosphere Monitoring Service reanalysis (CAMSRA), which is compared against in situ measurement."*

Line 105: the selection criterion for high-depletion and low-depletion years ("disrupted by SSW") should be further specified. Did you mean that any occurrence of a SSW (where? When?) is enough to call a year a low ozone depletion year? An objective criterion for "disrupted" is needed as well. Any ozone concentrations threshold used?

**Answer**: We added a new section (3.2) to describe how we distinguished low and high depletion years. Also, in Table 2, we summarized the high and low depletion years for each hemisphere:

***"3.2 High and low ozone depletion years definition***

*We distinguished low ozone depletion years from high ozone depletion years according to the occurrence of SSW events. Therefore, we applied the definition proposed by (Charlton and Polvani, 2007) to detect major SSW, which is based on determining the reversal of the daily-mean, zonal-mean zonal winds from westerly to easterly at 60°N and 10 hPa from November to April. **Table 2** shows the first day (central date) on which the daily zonal mean zonal wind at 10 hPa and 60°N changed from westerly to easterly between November and March. During the detection procedure, we required 20 consecutive days with westerly winds before identifying another event. We excluded cases with easterly zonal winds that did not return to westerly for at least 10 consecutive days before 30 April. In the SH, the shift from westerly to easterly is considered between July and October, and we excluded the cases when the wind did not return to westerly by 30 November."*

Table 1: see my general comment on a possible temporal mismatch of the NH and SH UTLS ozone observations.

**Answer**: We followed the recommendation to increase the analysis period. Table 1 describes the new periods and availability, which start in 2002.

**Table 1**: Summary of data used. The columns detail dataset sources, spatial coverage, temporal periods, the number of flights in the period and the number of measurements for each variable (in ozonesondes is the number of interpolated measurements) between 200-300 hPa.

| Data source | Hemisphere or site (lat., lon.) | Periods | No. flights or valid launches | Variable (No. measurements) |
|---|---|---|---|---|
| SouthTRAC | SH (45ºS-60ºS, 30ºW-85ºW) | 4 Sep - 20 Nov 2019 | 16 | Pressure: 43k $O_3$: 39k CO: 26k $H_2O$: 43k RH: 43k |
| WOUDC | 1. Ushuaia (54.85ºS, 68.31ºW), 2. Lauder (45.04ºS, 169.68ºE), 3. Macquarie (54.50ºS, 158.95ºE) | 1. 2008-2022 2. 2002-2022 3. 2002-2022 | 1. 141 2. 232 3. 203 | 1. Pressure: 7.1k $O_3$: 7.1k RH: 7.1k 2. Pressure: 12k $O_3$: 12k RH: 12k 3. Pressure: 11K $O_3$: 11k RH: 11k |
| IAGOS | NH (45°N-60°N) | 2002-2022 | 6,315 | Pressure: 16M $O_3$: 11M CO: 11M $H_2O$: 14M RH: 15M |

Lines 142-145: it should be clearly mentioned here that you describe the RH measurements of the radiosonde to which the ozonesonde is coupled for data transmission and auxiliary meteorological measurements (air pressure, temperature, relative humidity, wind direction and speed).

> **Answer**: Thank you for noticing this. We added: "*These ozonesondes were coupled with radiosondes, which measured pressure, temperature, RH, wind speed and wind direction.*"

Section 2.4: the IAGOS-CORE ozone measurements have not been described here.

> **Answer**: Please check again because it is described: "*IAGOS-CORE provides ozone (and carbon monoxide) data using an ultraviolet (infrared) absorption spectrometer, with an accuracy, precision and time response of 2 nmol mol$^{-1}$, 2% and 4 s (5 nmol mol$^{-1}$, 5%, 30 s) respectively.*"

Lines 167-168: the RH measurements are done by the Vaisala radiosonde sensors, not by the ozonesonde. So replace "ozonesonde" with "radiosonde" here (and at other locations).

> **Answer**: We appreciate noticing this, and we have changed this typo throughout the document.

Lines 179-180: "we reduced the number of pressure bins between 300 and 200 hPa to 5 bins": for CAMS or for the RS and IAGOS measurements? Not clear from the wording here.

> **Answer**: We removed this sentence from the description of "CAMS reanalysis" and added in section 3.3: "*In **Figure 6**, the ozone boxplot is displayed in five pressure bins, determined by the five vertical levels available for CAMS in 300-200 hPa.*"

Lines 182-184: is there a priority given to those two principles (better: criteria)?

**Answer:** We reworded the following: "*In the following sections, we mainly used relative humidity to assign the stratospheric air character in the UTLS. At the same time, in some specific events, we also take advantage of the enhanced trace gases simultaneously measured, at high time resolution, on board HALO to further characterize the tropospheric or stratospheric origin of the air masses.*"

Caption Fig. 2: add the LDY and HDY abbreviations after low and high-depletion years, as those abbreviations are used in the figure legend.

**Answer**: We made the change suggested.

Lines 203-204: "On this regard, we found CO mixing ratios up to 319 ppb between ...": This cannot be seen from the plot. Not clear if the highest CO values without simultaneous O3 measurements are meant here. Clarify.

**Answer**: In the manuscript's new version, the period is longer for IAGOS data (2002-2022), so we have eliminated that sentence and left the main message: *Notice that some CO values are not included in NH because they were not simultaneously measured with ozone.*

Line 219: "tracers" instead of "traces"

**Answer**: We modify the text as suggested.

Line 226: replace "ozonesondes" with radiosondes.

**Answer**: We modify the text as suggested.

Fig 5: specify in the discussion around this figure which High-depletion and low-depleting years have been used for both the NH and SH in the CAMS calculations. In Section 2.6, I think you mention that the same years as for the observations (Table 1) are used, which would be a pity, given the longer available time range for CAMS. Could you provide the same figure as Fig. 5, but now for the entire available CAMS time range, to investigate the temporal impact on the comparisons? That would be a nice plus.

**Answer**: In the new section: 4.3 Outputs comparison based on in situ measurement and reanalysis, we introduce Figure 6 (before Fig. 5) with the following line: *Therefore, the motivation of this section is to compare the in situ measurements of ozone, filtered by 20% RH in the UTLS, with the values produced by the CAMS reanalysis over the entire longitudinal band within 45-60° latitude and for low and high depletion years indicated in **Table 2**, aiming to provide a spatially resolved perspective.*

**Table 2** indicates the low and high depletion years utilized for Figures 5 and 6. Notice that by extending the period, we now have two SSW events in the SH.

**Table 2**: Study period considered for both hemispheres, subperiods and low ozone depletion years according to SSW definition.

| Study period | Late winter-early spring (no. flights or launches) | Mid spring (no. flights or launches) | Low-depletion years (central date) |
|---|---|---|---|
| 4 Mar - 20 May | 4 - 31 Mar (456 flights) | 1 Apr – 20 May (835 flights) | 2002 (17 Feb), 2003 (18 Jan), 2004 (5 Jan), 2006 (21 Jan), 2007 (24 Feb), 2008 (22 Feb), 2009 (24 Jan), 2010 (24 Mar), 2013 (6 Jan), 2018 (12 Feb), 2019 (1 Jan) and 2021 (5 Jan) |
| 4 Sep - 20 Nov | 4 - 30 Sep (10 flights) (204 launches) | 1 Oct - 20 Nov (6 flights) (372 launches) | 2002 (25 Sep) and 2019[†] (15 Sep) |

[†]: Central date determined when the zonal-mean zonal winds at 60°S and 10 hPa decrease to $\leq 20$ m/s (Rao et al., 2020).

Fig 5: why do you have lower observation numbers in the pressure bins centered around 270 hPa?

**Answer**: The answer to this question involves aircraft companies decisions which uses specific flight altitudes in some regions. Particularly, in the North Atlantic region (50 - 20W, 50 - 60N), the 270 hPa bin was not sampled during March-May, contrary to the other regions.

Lines 280-283: "We delve into the apparent overestimation of ozone vertical gradient and medians obtained from the CAMS reanalysis across the entire longitudinal band of the Southern Hemisphere for high depletion years compared to ozonesonde measurements from the period 2008-2018 (Fig. 6)": I really do not have a clue where all these findings come from. Where do I see the apparent overestimation of ozone vertical gradient and medians? This can only be in Fig.5, I assume, but then late winter – early spring should be specified. But CAMS has not been used for the entire period 2008-2018? And how can you extend this finding for the ozone vertical gradient to the entire longitudinal band of the SH with Fig. 6? Not clear at all.

**Answer**: The new version of the manuscript benefits from the longer period and the ozonesondes launched at Lauder and Macquarie Island in the SH. These two changes eliminate the overestimation we faced with fewer data. In the new **section 4.3**, we described the new Figure 6 (previously Figure 5) in two terms: the better agreement between CAMS and the SouthTRAC period (the one with the higher frequency of measurement) and the improvement for the entire period when considering ozonesondes from Ushuaia, Lauder and Macquarie Island. We have selected some lines from the **section 3.3**:

*"**Figure 6** also clearly illustrates the similarity between the ozone medians obtained from the CAMS reanalysis and SouthTRAC in the period with the highest number of flights, i.e., late winter-early spring, for pressures below 270 hPa."*

And:

*"Hence, considering only ozonesondes, the enhancement for mid-spring was 17% (33 $\pm$ 10 nmol mol$^{-1}$) and for the entire period (SouthTRAC combined with ozonesondes) it was 24% (43 $\pm$ 13 nmol mol$^{-1}$)."*

Note we have also added a new **Figure B1** in Appendix B, which describes the Ozone boxplot without SouthTRAC data (also arising in the general comments by the reviewer)

Lines 286-288: should be clarified and quantified. How much is this slight improvement or worsening? At which pressure bins?

**Answer**: We eliminated those lines and Appendix C because, as indicated in the previous answer, we have improved our analysis with more ozonesonde and a longer period.

Fig. 6: Mark the SouthTRAC measurement region on the SH map here. Specify for which vertical range the ozone concentrations are shown.

**Answer**: In our opinion, it is a bit redundant to indicate the SouthTRAC measurement region again since it is described in **Figure 1**. On the other hand, we added in the caption of **Figure 7** (previously Fig. 6): *"...between 300-200 hPa"*.

Start the conclusions with some lines describing the SouthTRAC observations.

**Answer**: We added: *" The SouthTRAC mission represents an invaluable contribution to atmospheric chemistry research in the Southern Hemisphere. This aircraft campaign provided unique data recorded during a rare stratospheric sudden warming event in the Southern Hemisphere that produced profound effects on the chemical structure of the atmosphere, leveraged in this research as a proxy for a diminished ozone depletion scenario. The main conclusions of our research are listed below."*

Line 299: add 2019 after 12 Nov.

**Answer**: We modify the text as suggested.

Lines 299-302: this paragraph in the conclusions came a bit like a surprise, and I assume you are deepening a bit the analysis described in lines 215-221. If this finding is so important that an entire paragraph in the conclusions, with concentration numbers, is devoted to it, this would deserve also more details in the main manuscript as well (and the authors might consider moving the plot from the appendix to the main manuscript).

**Answer**: We agreed, so we moved the Figure to the main text (now **Figure 4**). We also added in the main text: *"Ultimately, the impact of emissions from the Australian bushfires in the UTLS may be interpreted as feedback initiated early in August-September 2019, which derived from the deceleration of the polar vortex in the middle stratosphere."*

Lines 310-314: isn't this a too strong conclusion, based on a very small number of statistics (i.e. low number of high-ozone depleting years in NH and low number of low-ozone depleting years in SH)? And in the NH you consider the entire latitudinal band, and only a small zone in the SH latitudinal band. You wrote this yourself in lines 321-322.

**Answer**: After the improvements included in this new version, we reworded this conclusion a bit: *"The interannual ozone variability in the NH UTLS (March-May) as a function of SSW events resulted in a 9% ozone enhancement ($31 \pm 11$ nmol mol$^{-1}$) compared with years without SSW. This enhancement was nearly three times larger in the SH, reaching 24% ($43 \pm 13$ nmol mol$^{-1}$)."*

**General Comments:**

The text is polished and the figures are easy to digest and well-constructed. The SouthTRAC September-November 2019 aircraft data are serendipitous in that you can examine a rare Antarctic SSW and low depletion event with multiple in-situ chemical species (including some aged fire plumes). However, this paper is missing a lot of the broader context that would be provided by analysis of the Lauder (45S) and Macquarie Island (55S) ozonesonde records. Both of those records predate 2000 and would enable you to also examine the 2002 Antarctic SSW event, effectively doubling your SH low-depletion cases. Those two stations are on nearly the complete opposite side of the globe, adding longitudinal contrasts that are important, as you show in Figure 6. In fact, you might end up with different results with the inclusion of more SH data and a second SSW year in 2002. The SouthTRAC and Ushuaia ozone data are close to a local minimum in ozone according to Figures 1 and 6. In either case, you will add confidence to your results.

The study periods for the NH and SH should be aligned, something like 2002-2022 for both hemispheres, in my opinion. There should be plenty of IAGOS data in the NH to extend the study, and of course there is a dense network of ozonesonde stations in Europe and North America in your 45-60N band of interest. At the very least, why not align the study periods for both hemispheres and include the 2019 Ushuaia data?

Finally, I am curious to know who oversees the Ushuaia station and if they were offered co-authorship. Often, this recognition can help maintain a station's support.

In my assessment, this paper is not ready for publication and needs major revisions. The addition and analysis of more datasets, alignment of the study periods in both hemispheres, and a broader context and discussion of the impact of the differing STE ozone amounts between the two hemispheres will make this an important contribution.

> **Answer**: We will address all the aspects mentioned above in the specific comments. However, at this point, we indicate that we extended the analysis period and included two additional ozonesondes launched within the latitudinal band of interest. This improvement allowed a broader discussion and added greater confidence to our results. In summary, the new periods analyzed are:
>
> - IAGOS: 2002-2022
> - Ozonsonde launched from Lauder and Macquarie Island: 2002-2022
> - CAMSRA: 2003-2022

**Specific comments:**

Abstract lines 15-17: The answer to the question raised in the first sentence of the Abstract is well known and does not accurately capture what I think you are trying to demonstrate in this manuscript: Even during low depletion SH years, Southern Hemisphere UTLS ozone in STE events is far less than Northern Hemisphere high depletion years. The NH vs. SH lower stratospheric ozone differences in these latitude bands are obvious from the global ozonesonde network. See my quick analysis in Figure R1:

Abstract Line 26: "…the SSW event increased SH UTLS **ozone** by 37%..."

> **Answer**: We appreciate this comment since it permits us to better focus our research. Accordingly, we removed those lines (15-17) and added: *"Ozone changes in the upper troposphere-lower stratosphere (UTLS) resulting from dynamical and chemical processes strongly affect the atmosphere's radiative forcing. This study analyzed intra- and interhemispheric ozone differences in the UTLS within the 45-60° latitude band, distinguishing between years disrupted by sudden stratospheric warming (SSW) events from 2002 to 2022."*
>
> Regarding line 26, we emphasize that the work benefits greatly from the incorporation of the suggestions of both reviewers and therefore we have modified the last line of the summary due to the confidence gained: *"Notably, the SSW events (2002 and 2008) increased SH UTLS ozone by 24% (43 nmol mol$^{-1}$) compared to high depletion years, while in the NH, the increase was 9% (31 nmol mol$^{-1}$)."*

Lines 82-83: Again, we can refer to Figure R1 to demonstrate that SH STE events will contain less ozone than NH STE events. There is simply less ozone in the SH mid-latitudes.

> **Answer**: We thank you pointed this out. We think the following lines better reflect our *research: "Ozone in the UTLS is highly variable, driven by a complex interplay of dynamical and chemical processes (Millán et al., 2024; Bourgeois et al., 2020; (Neu et al., 2014); Riese et al., 2012). In this context, SSW events represent a diminished ozone depletion scenario, which, combined with the highly resolved profiles obtained during the SouthTRAC campaign, provides a unique and realistic framework for assessing ozone changes…"*

Line 97: It is contained in Table 1, but please state in the main text what time periods you are focusing on. Note also my concerns about mismatched time periods for the NH and SH.

> **Answer**: We added the new period in the main text of Section 3.1: *"For comparison, IAGOS data from 2002 to 2022…"*

Line 104: "…interannual variability of both hemispheres…"

> **Answer**: We modify the text as suggested.

Table 1: Why are the time periods examined not the same for NH and SH? Surely there are enough NH IAGOS data to make the analysis period for both hemispheres 2008-2022 or so? Also, extending back to 2002 will enable you to add a second SH SSW event. By adding Lauder and Macquarie Island ozonesonde data to the analysis, this can easily be accomplished for the Southern Hemisphere as well.

> **Answer**: We extended the time back to 2002 and added Lauder and Macquarie Island. All this info is summarized in **Table 1**:

**Table 1**: Summary of data used. The columns detail dataset sources, spatial coverage, temporal periods, the number of flights in the period and the number of measurements for each variable (in ozonesondes is the number of interpolated measurements) between 200-300 hPa.

| Data source | Hemisphere or site (lat., lon.) | Periods | No. flights or valid launches | Variable (No. measurements) |
|---|---|---|---|---|
| SouthTRAC | SH (45ºS-60ºS, 30ºW-85ºW) | 4 Sep - 20 Nov 2019 | 16 | Pressure: 43k
$O_3$: 39k
CO: 26k
$H_2O$: 43k
RH: 43k |
| WOUDC | 4. Ushuaia (54.85ºS, 68.31ºW),
5. Lauder (45.04ºS, 169.68ºE),
6. Macquarie (54.50ºS, 158.95ºE) | 4. 2008-2022
5. 2002-2022
6. 2002-2022 | 1. 141
2. 232
3. 203 | 4. Pressure: 7.1k
$O_3$: 7.1k
RH: 7.1k
5. Pressure: 12k
$O_3$: 12k
RH: 12k
6. Pressure: 11K
$O_3$: 11k
RH: 11k |
| IAGOS | NH (45°N-60°N) | 2002-2022 | 6,315 | Pressure: 16M
$O_3$: 11M
CO: 11M
$H_2O$: 14M
RH: 15M |

Line 142 and other locations: RS92 radiosondes

**Answer**: We appreciate noticing this, and we have changed this typo throughout the document.

Lines 167 and 168: The RH measurements come from the Vaisala radiosondes, not the ozonesonde instrument

**Answer**: We modify the text as suggested.

Figure 2: It looks to me like the water vapor measurements are rounded to the nearest whole µmol mol-1, but are there really measurements of just 1 µmol mol-1?

**Answer**: Below 10% RH, the measurement uncertainty with the IAGOS-Core instruments becomes as high as the measured value itself. Also, the well-visible vertical $H_2O$ lines are due to the fact that the IAGOS-Core $H_2O$ mixing ratios are available as integers in the database. Given that, we preferred to remove water values below 7 ppmv as it does not affect the main message we want to convey.

Line 219: "tracers" not "traces"

**Answer**: We modify the text as suggested.

Line 226: Again, the RH measurements are from the radiosondes attached to the ozonesondes

**Answer**: We made the change suggested.

Line 232: "…applies to the 230-220 hPa bin…"

**Answer**: We made the change suggested

Figure 4: Why did you not include 2019 Ushuaia data in addition to SouthTRAC?

**Answer**: **Figure 5** (previously Fig. 4) includes the ozonesondes (indicated in **Table 1**) and SouthTRAC data. We also added in Appendix B the same analysis without SouthTRAC.

Line 242: Are these differences statistically significant? It might be difficult to draw conclusions based on the fairly limited data from single SH low depletion case in 2019.

**Answer**: Our analysis was substantially improved by extending the period analyzed. Line 241-242 was replaced by: "*In the NH, interannual ozone variability increased by 9% (31 ± 11 nmol mol$^{-1}$) compared with high depletion years. The changes in the subperiods late winter-early spring and mid-spring were similar, with 8% (26 ± 22 nmol mol$^{-1}$) and 10% (36 ± 12 nmol mol$^{-1}$), respectively.*"

Figure 6: Is the UTLS here also defined as 300-200 hPa? Please state clearly in the Figure 6 caption or plots

Answer: Yes, in **Figure 7** (previously **Fig. 6**) the pressure range is 300-200 hPa. Here is the caption: "*Figure 7: Ozone medians (nmol mol$^{-1}$) obtained from CAMS reanalysis filtered by RH lower less 20% at a resolution of 0.75° x 0.75° and between 300-200 hPa. The upper panel depicts the ozone median in UTLS of the northern hemisphere, separated into low and high depletion years (entire period), while the lower panel describes the same for the southern hemisphere. Note that the color scales are different in the NH and SH given the strong differences in ozone mixing ratios.*"

Figure 6 Caption: "…the same for the Southern Hemisphere"

**Answer**: We deleted the "s". We appreciate it.

Line 287-288: This sentence is unclear. Please rewrite.

**Answer**: We eliminated this sentence as well as Appendix C.

Lines 321-323: There is certainly a lack of in-situ ozone profile data in the SH, but you are not taking full advantage of what is available in this latitude band, including the Lauder and Macquarie Island ozonesonde datasets

**Answer**: We appreciate this suggestion, which let us improve our analysis substantially.

**Community Comments (CC) by** Keding Lu

**General Comments:**

The differences between the NH and SH mid-latitude UTLS ozone concentrations are of high interest for both the global radiative forcing and the global ozone budget studies. This paper provide timely analysis on a super valuable dataset on the mid-latitude SH UTLS measurements which could open a window of this demanding topic.

This manuscript has already received two thorough and critical reviews from two anonymous referees regarding the scope, scientific analysis and structure of the paper and I don't have anything to add. Nevertheless, I would agree with the two reviewers that the significant different sample size could be a point to further address of which in this case the two ozone sonde observations in SH could be considered.

Overall, for my responsibility, I just found no discrepancies between the conclusions and the findings of other papers submitted so far to the TOAR-II Community Special Issue.

**Technical comments**

1. The term 'ozone' is sometimes depicted as '$O_3$' in both text and figures, maybe the authors can choose '$O_3$' in all the figures while 'ozone' in the text, in addition, the font style might be synchronized too.

> **Answer**: We appreciate the comments and also agree with the reviewers on the benefits of increasing the analysis period and adding ozone soundings available at the SH.

> On the other hand, we used "ozone" instead of "$O_3$" throughout the text except in Tables and Figures. The same applies to carbon monoxide (instead of CO).

---

## Referee Report (RR1)

**Review of "Hemispheric differences in ozone across the stratosphere-troposphere exchange region" by Sequel et al.**

The manuscript compares ozone mixing ratios of stratospheric ozone in the UTLS between the northern and southern hemisphere mid-latitudes, and between high and low ozone depletion years. The analysis is mainly based on in-situ measurements (aircraft and ozonesondes), but is complemented with CAMSRA model output for a better spatiotemporal representation.

I first want to thank the authors for following my suggestion to extend the time range of their original study period and to incorporate additional SH ozonesonde sites (Lauder and Macquarie Island). The manuscript is now also better organized and most of my questions for clarifications have been answered.

**General comments**

However, I'm still missing somewhat the purpose and focus of the paper. The purpose of the paper and the method followed by achieving this should be mentioned more clearly in the beginning of the manuscript. You describe some different elements of the puzzle in each paragraph of the introduction (STE, available data, SSW), but you do not lay the puzzle yourself by e.g. explicitly linking these phenomena (STE, SSW) with their possible impact on hemispheric ozone differences, why concentrating on mid-latitudes in this study, why it is important to discriminate between high and low ozone depleting years in this study. You assume that the reader will see the puzzle during the course of the paper. Also in the introduction, I'm missing some insight on what the manuscript wants to add to the current knowledge?

Therefore, 2 suggestions, for the introduction:
- add what is already known about UTLS ozone mixing ratio differences between the NH and SH, and what is known already about the impact of high/low ozone depletion in the stratosphere on UTLS ozone.
- Add a true roadmap for your study and give a short explanation for each step (instead of the paragraph from lines 85 to 96), for instance:
  - **We want to study hemispheric ozone differences in the UTLS at mid-latitudes during spring**. Why in the UTLS? Why at mid-latitudes? Why during springtime? Why are such possible hemispheric ozone differences important?
  - **We will only look at the UTLS ozone of stratospheric origin**. Therefore, in our analysis, we make a distinction between high and low stratospheric ozone depleting events/years.

**Specific comments**

- After 2 Data and before 2.1 SouthTRAC data, write a short introductory paragraph in the style of "The UTLS ozone measurements used in this study are available from (research + commercial) aircraft and ozonesondes and are complemented with chemical reanalysis vertical ozone

profiles. To determine the stratospheric or tropospheric origin of the ozone data, we used water vapour (or humidity) measurements from the aircraft and radiosondes coupled to the ozonesondes and from the chemical reanalysis, and additionally CO, $HNO_3$, HCl measurement from some of the aircraft data. In the next section, we give more details on these datasets.

- Caption Fig. 1: mention explicitly for which time period the number of observations is shown here.
- Again, after 3 Method and before 3.1 Study period & UTLS definition, it is important to provide some guidance to the reader. Therefore, write a short introductory paragraph in the style of "With the data available and described in the previous section, we will now describe how we will analyze springtime UTLS ozone differences at the mid-latitudes between both hemispheres. We first describe how our study period and the used UTLS definition in 3.1, mention how we distinguish between high and low stratospheric ozone depletion years in 3.2, and we show how we ascertain the stratospheric origin of the analyzed UTLS ozone concentrations in 3.3."
- Section 3.1 misses a real focus. The first two lines (176-177) belong to the SouthTRAC data description. Define the UTLS (300-200 hPa) and the free troposphere (700-300 hPa) clearly. Mention that your study focus on the springtime only, and already define this periods (4 Mar – 20 May, NH & 4 Sep – 20 Nov, SH) here.
- Section 3.2 and Table 2: Here, some major clarifications are needed. Low ozone depletion years are defined by the presence of a SSW event (last column of Table 2). However, for most NH years, the SSW central date lies well ahead of the study period (Mar – May), so it is not clear of the SSW event still occurs during the study period. If this is not the case, it should be mentioned what the expected impact of a SSW event earlier that year would be on the UTLS springtime ozone concentrations. To me, a more direct distinction between high and low ozone depletion years could be made by simply looking at the stratospheric (or total) ozone amounts, averaged over the 45-60° latitude bands, for the 4 Mar – 20 May (NH) or 4 Sep – 20 Nov (SH) period. Please discuss.
- Table 2: add a column with NH and SH to the left, add either "IAGOS" or "SouthTRAC" before flights and add that these latter flights only occurred in 2019. Also, I don't understand why the total number of IAGOS flights in the NH for the two periods (later winter-early spring & mid spring), resp. 456 and 835 flights, is not equal to the number of IAGOS flights in Table 1 (6315).
- To me, it makes more sense to incorporate parts of section 4.1 in section 3.2. In its current form, section 4.1 gives the impression of being a rather standalone section, and does not entirely seem to fit within the logical flow of the paper. You could solve this by transferring parts to section 3.2. Basically, in section 3.2, you want to isolate the air of stratospheric origin in the UTLS to look at its ozone concentration properties. So section 4.1 should make a link to this section, but now using tracer correlations to look at the origin of the UTLS air masses. Also the choice of the 4.1 section title could be better. The sentences at the end of page 8 might be replaced by a better guidance to the analyses done in sections 4.1 and 4.2, and referring more directly to these sections (rather than describing in rather general terms, as is done now).
- Figure 2: in addition to my previous comment: argue why these O3-H2O tracer correlations are important, and what you learn from them, and if there is difference between the LD and HD correlations. Also, specify in the caption which periods (2002-2022?, 4 Mar – 20 May NH?, 4 Sep – 20 Nov SH?) have been used for the correlation plots.

- Figure 3: specify in the caption which periods (2002-2022?, 4 Mar – 20 May NH?, 4 Sep – 20 Nov SH?) have been used for the correlation plots.
- Figure 4: specify in the caption that these are SouthTRAC flights only!
- Line 269: what does the comparison between Fig. 5 (stratospheric origin) and A1 (mixture of tropospheric and tropospheric origin) learn us? For instance, for late winter – early spring the SH LD ozone amounts are much larger than the SH HD ozone amounts in A1 (no RH filter), compared to the same comparison in Fig. 5 (RH filter). Any clue for this?
- On page 12, line 280, you define differences between high and low-depletion years as "interannual variability". I don't think that "interannual variability" is a good term for it; it is such a general term. Also you should define how you calculate the ozone difference between high and low-depletion years. It is simply the difference between the mean of the overall springtime ozone between 300-200 hPa for both the LD and HD years, or do you somewhat average out the mean values for every pressure level, as shown in Fig. 5? How are then the values mentioned in lines 280-282 calculated? Please specify! From Fig. 5, I would assume that the ozone differences between LD and HD years are nowhere significant, so I don't understand quite well how the values, and their uncertainties, in lines 280-282 have been obtained.
- In Fig. 6: can't the comparisons (vertical dashed lines) with the ozone medians in the NH obtained from IAGOS added to this figure, similarly as has been done for the SH?
- Related to previous comment: the analysis of this figure 6 is only used for comparing the SouthTRAC measurements (2019 only) with the CAMS reanalysis output (the low-depletion years 2002 and 2019?). Is this the most important message from this Figure? Shouldn't the (pattern) agreement between Fig. 5 (observations) and Fig. 6 (model) be discussed first? E.g. the higher NH ozone values, the slightly (but not significant) higher amounts for LD years compared to HD years? And how do the model values relate with the measurement values (for all measurement types, and in NH and SH)?
- Lines 314-325: those findings are really not obvious at all from the figure. Are you saying that the agreement between CAMS and SouthTRAC (SH LD, light blue) is better during late winter – early spring compared to mid spring? On which ground? For pressures lower than 270 hPa (instead of below 270 hPa)? What do you mean with higher "fluctuations" in the medians for the mid-spring period? In the vertical or looking at the percentiles of the medians (I don't see this for the latter)? Are fluctuations in the vertical relevant given the coarser vertical pressure grid for CAMS?
- Figure 7: in the discussion of Fig. 7, it might be informative to give the overall mean UTLS ozone value for each considered case (NH LD, NH HD, SH LD, SH HD).
- Conclusions: if you put so much weight on the analysis in section 4.1 (two paragraphs devoted to it in the conclusions), you should stress much more the importance of this analysis in the main text as well. As written before, for me, it really hampers the logical flow of the paper a bit in its present form.

**Technical corrections**
- Line 39: remove the dot before OH
- Line 86: remove the brackets before and after Neu et al., 2014
- Line 86: remove "diminished"
- Line 90: remove "from"

- Line 93: in situ measurementS. Futher: replace "air masses exhibiting stratospheric character" with "air masses of stratospheric origin"
- Line 135: replace so-called initiative with the Tropospheric Ozone Assessment Report – Phase II (TOAR-II) Focus Working Group HEGIFTOM (Harmonization and Evaluation of Ground-Based Instruments for Free-Tropospheric Ozone Measurements).
- Line 161: an identical instead of the identical
- Line 184: proposed by Charlton and Polvani (2007)
- Line 207: Section 4.1 instead of Section 3.2
- Line 237: give the pressure level instead of "above 9 km".
- Line 266: and instead of y
- Line 305: add ozone between percentage and difference
- Line 358: replace "characterized by" with "of".

---

## Author Response (AR2)

**Author Comments (egusphere-2024-3719)**

Manuscript title: Hemispheric differences in ozone across the stratosphere-troposphere exchange region

**Referee #1**

The manuscript compares ozone mixing ratios of stratospheric ozone in the UTLS between the northern and southern hemisphere mid-latitudes, and between high and low ozone depletion years. The analysis is mainly based on in-situ measurements (aircraft and ozonesondes), but is complemented with CAMSRA model output for a better spatiotemporal representation.

I first want to thank the authors for following my suggestion to extend the time range of their original study period and to incorporate additional SH ozonesonde sites (Lauder and Macquarie Island). The manuscript is now also better organized and most of my questions for clarifications have been answered.

**General comments:**

However, I'm still missing somewhat the purpose and focus of the paper. The purpose of the paper and the method followed by achieving this should be mentioned more clearly in the beginning of the manuscript. You describe some different elements of the puzzle in each paragraph of the introduction (STE, available data, SSW), but you do not lay the puzzle yourself by e.g. explicitly linking these phenomena (STE, SSW) with their possible impact on hemispheric ozone differences, why concentrating on mid-latitudes in this study, why it is important to discriminate between high and low ozone depleting years in this study. You assume that the reader will see the puzzle during the course of the paper. Also in the introduction, I'm missing some insight on what the manuscript wants to add to the current knowledge?

Therefore, 2 suggestions, for the introduction:

- add what is already known about UTLS ozone mixing ratio differences between the NH and SH, and what is known already about the impact of high/low ozone depletion in the stratosphere on UTLS ozone.

- Add a true roadmap for your study and give a short explanation for each step (instead of the paragraph from lines 85 to 96), for instance:

  - We want to study hemispheric ozone differences in the UTLS at mid-latitudes during spring. Why in the UTLS? Why at mid-latitudes? Why during springtime? Why are such possible hemispheric ozone differences important?

  - We will only look at the UTLS ozone of stratospheric origin. Therefore, in our analysis, we make a distinction between high and low stratospheric ozone depleting events/years.

**Answer**: We appreciate the suggestions, and we modified the last paragraph of the introduction accordingly. The text: *"Given the multiple dynamical and chemical processes that influence ozone in the UTLS (Millán et al., 2024; Bourgeois et al., 2020; Neu et al., 2014; Riese et al., 2012) and the interhemispheric differences in processes such as the magnitude of the stratospheric ozone depletion and frequency of SSW events, this study, investigates hemispheric differences in UTLS ozone in air masses of*

*stratospheric origin, with focus on mid-latitudes, where STE plays a dominant role in determining ozone levels. We also leverage the increased ozone abundance under low-depletion conditions derived from SSW events to determine the intrahemispheric UTLS ozone differences. Through this comparative analysis, we aim to quantify the influence of two processes, i.e., stratospheric ozone depletion and SSW, on the ozone UTLS, where this species is an important radiative forcer. Our analysis utilized stratospheric and tropospheric chemical tracers measured during the SouthTRAC mission, by IAGOS commercial aircraft and by ozonesondes. Spatial coverage was further enhanced using the Copernicus Atmosphere Monitoring Service reanalysis (CAMSRA), which we compared against in situ measurements..."*

Regarding the point suggesting adding what is known about hemispheric differences to the best of our knowledge, we have included relevant references to build our case. However, we apologize in advance if we are missing some key papers or review papers.

**Specific comments:**

▪ After 2 Data and before 2.1 SouthTRAC data, write a short introductory paragraph in the style of "The UTLS ozone measurements used in this study are available from (research + commercial) aircraft and ozonesondes and are complemented with chemical reanalysis vertical ozoneprofiles. To determine the stratospheric or tropospheric origin of the ozone data, we used water vapour (or humidity) measurements from the aircraft and radiosondes coupled to the ozonesondes and from the chemical reanalysis, and additionally CO, HNO3, HCl measurement from some of the aircraft data. In the next section, we give more details on these datasets.

**Answer**: We appreciate the suggestion and accordingly we added in the main text: *"The UTLS ozone measurements used in this study are available from (research & commercial) aircraft and ozonesondes. These in situ measurements were complemented with vertical ozone profiles obtained from chemical reanalysis. To determine the stratospheric or tropospheric origin of the ozone data, we used water vapor (or humidity) measurements from the aircraft and radiosondes coupled to the ozonesondes and from the chemical reanalysis in addition to carbon monoxide (CO), nitric acid ($HNO_3$) and hydrogen chloride (HCl) measurements from some of the aircraft data. In the next sections, we provide more details on these datasets".*

▪ Caption Fig. 1: mention explicitly for which time period the number of observations is shown here.

**Answer**: We added: *"SouthTRAC shows data from 4 Sep to 20 Nov 2019 and IAGOS between 4 Mar and 20 May from 2002 to 2022."*

▪ Again, after 3 Method and before 3.1 Study period & UTLS definition, it is important to provide some guidance to the reader. Therefore, write a short introductory paragraph in the style of "With the data available and described in the previous section, we will now describe how we will analyze springtime UTLS ozone differences at the mid-latitudes between both hemispheres. We first describe how our study period and the used UTLS definition in 3.1, mention how we distinguish between high and low stratospheric ozone depletion years in 3.2, and we show how we ascertain the stratospheric origin of the analyzed UTLS ozone concentrations in 3.3."

**Answer**: We added: *"In this section we describe how we used the data available to analyze springtime UTLS ozone differences at the mid-latitudes between both hemispheres. We first describe our study period and the used UTLS definition (section 3.1), then we mention how we distinguish between high and low stratospheric ozone depletion years (section 3.2) and show how we ascertain the stratospheric origin of the analyzed UTLS ozone mixing ratios (section 3.3)."*

▪ Section 3.1 misses a real focus. The first two lines (176-177) belong to the SouthTRAC data description. Define the UTLS (300-200 hPa) and the free troposphere (700-300 hPa) clearly. Mention that your study focus on the springtime only, and already define this periods (4 Mar –20 May, NH & 4 Sep – 20 Nov, SH) here.

**Answer**: We modified the beginning of 3.1 as follows: *In this study, we denominate the UTLS as the segment between 200 (~12 km) and 300 (~9 km) hPa, while the free troposphere is the segment between 700 to 300 hPa. We focus on the period between late-winter and mid-spring in both hemispheres: 4 Sep–20 Nov (SH) and 4 Mar–20 May (NH).*

▪ Section 3.2 and Table 2: Here, some major clarifications are needed. Low ozone depletion years are defined by the presence of a SSW event (last column of Table 2). However, for most NH years, the SSW central date lies well ahead of the study period (Mar – May), so it is not clear of the SSW event still occurs during the study period. If this is not the case, it should be mentioned what the expected impact of a SSW event earlier that year would be on the UTLS springtime ozone concentrations. To me, a more direct distinction between high and low ozone depletion years could be made by simply looking at the stratospheric (or total) ozone amounts, averaged over the 45-60° latitude bands, for the 4 Mar – 20 May (NH) or 4 Sep – 20 Nov (SH) period.

**Answer**: In the first revision, we were asked to include an objective criterion for the selection of high and low ozone depletion years, so we decided to choose the most commonly used metric to detect SSWs. In this metric, the central (onset) date is the first day on which the daily mean zonal mean zonal wind at 60°N and 10 hPa is easterly. Despite the metric to detect SSWs, in the NH, SSWs lead to higher ozone in springtime. See for, example, Figure 7 from *Bouillon, M., Safieddine, S., & Clerbaux, C. (2023). Sudden stratospheric warmings in the Northern Hemisphere observed with IASI. Journal of Geophysical Research: Atmospheres, 128, e2023JD038692.* https://doi.org/10.1029/2023JD038692).

[Figure]

**Figure 7.** Daily average total ozone column during the eight major SSW winters (dark green, with the standard deviation shown around it) compared to average of all winters from 2007–2008 to 2020–2021 (dashed light green). The daily average temperature at 10 hPa is shown in red.

- Table 2: add a column with NH and SH to the left, add either "IAGOS" or "SouthTRAC" before flights and add that these latter flights only occurred in 2019. Also, I don't understand why the total number of IAGOS flights in the NH for the two periods (later winter-early spring & midspring), resp. 456 and 835 flights, is not equal to the number of IAGOS flights in Table 1 (6315).

**Answer**: We added the corresponding database to the number of flights. We also corrected the number of IAGOS flights and now the total is 6315 flights, as indicated in Table 1. Regarding adding new columns, we prefer not to do that since the information is sufficiently clear. We also believe it is redundant to indicate in the table that SouthTRAC occurred in 2019 as this has been clearly stated in the methodology and other sections.

- To me, it makes more sense to incorporate parts of section 4.1 in section 3.2. In its current form, section 4.1 gives the impression of being a rather standalone section, and does not entirely seem to fit within the logical flow of the paper. You could solve this by transferring parts to section 3.2. Basically, in section 3.2, you want to isolate the air of stratospheric origin in the UTLS to look at its ozone concentration properties. So section 4.1 should make a link to this section, but now using tracer correlations to look at the origin of the UTLS air masses.

**Answer**: These are good suggestions, and we continue to develop them in the following two points. We moved part of the first text (in section 4.1) to section 3.3 (methodology) where better fits and rephrased: "*Stratospheric and tropospheric tracer scatterplots provide a magnitude of bidirectional exchange across the tropopause by the identification of stratospheric and tropospheric branches and mixed air regions according to their tracer abundances (Gettelman et al., 2011). Hence, in this study, our first analysis used tracer correlations to compare the northern (IAGOS) and southern hemisphere (SouthTRAC) mid-latitudes, and between high and low ozone depletion years (NH).*"

- Also the choice of the 4.1 section title could be better. The sentences at the end of page 8 might be replaced by a better guidance to the analyses done in sections 4.1 and 4.2,

and referring more directly to these sections (rather than describing in rather general terms, as is done now).

Answer: We changed the section title to "*Tropospheric and stratospheric tracer correlations*". We also reworded the beginning of the section: "*In the following section, we used tracer correlations to characterize the tropospheric or stratospheric origin of the air masses in the UTLS and the free troposphere of both hemispheres. In section 4.2, we applied filters mainly based on the relative humidity to assign the stratospheric origin in the UTLS and then determine the inter- and intrahemispheric ozone differences. Finally, in section 4.3, we compare the results based on situ ozone measurements with CAMSRA outputs*".

▪ Figure 2: in addition to my previous comment: argue why these O3-H2O tracer correlations are important, and what you learn from them, and if there is difference between the LD and HD correlations. Also, specify in the caption which periods (2002-2022?, 4 Mar – 20 May NH?, 4 Sep– 20 Nov SH?) have been used for the correlation plots.

**Answer**: We appreciate this comment and the intention to improve our work. Therefore, we added the following lines to ensure a better transition between sections 4.2 and 4.2: "*Overall, in this section, we found that the NH does not show differences between high and low depletion years. Therefore, in the following section, we test different filters to isolate the air of stratospheric origin in the UTLS and thus investigate its ozone mixing ratios during high and low depletion years in both hemispheres. In this section, we also showed that the NH has higher carbon monoxide levels…*"

We also added the exact periods in the caption of Figures 2 and 3.

▪ Figure 3: specify in the caption which periods (2002-2022?, 4 Mar – 20 May NH?, 4 Sep – 20 Nov SH?) have been used for the correlation plots.

**Answer**: We added: "…*The left panels show scatter plots for the period between 4 Mar and 20 May from 2002 to 2022 for the NH between 300-200 and 700-300 hPa (IAGOS data set described in Table 1). The right panels show the same for the SH between 4 Sep and 20 Nov 2019 (SouthTRAC campaign data described in Table 1)…*"

▪ Figure 4: specify in the caption that these are SouthTRAC flights only!

**Answer**: We added: "*…These scatter plots are based exclusively on SouthTRAC data…*"

▪ Line 269: what does the comparison between Fig. 5 (stratospheric origin) and A1 (mixture of tropospheric and tropospheric origin) learn us? For instance, for late winter – early spring the SH LD ozone amounts are much larger than the SH HD ozone amounts in A1 (no RH filter), compared to the same comparison in Fig. 5 (RH filter). Any clue for this?

**Answer**: The filter's main function is to eliminate levels with more tropospheric (or mixed) properties. This can be clearly seen in the figure below, where the lowest ozone levels tend to be eliminated after the filter. We've added the following line to strengthen the message: "*The filter's applicability can be clearly verified for the subperiod late winter-early spring in the SH, where differences between high and low ozone depletion years can be noticed without any filter (Figure A1). However, we can also note that the 20% RH filter is very effective at removing the lowest ozone mixing ratios with tropospheric origin (or mixed air).*"

[Figure]

- On page 12, line 280, you define differences between high and low-depletion years as "interannual variability". I don't think that "interannual variability" is a good term for it; it is such a general term. Also you should define how you calculate the ozone difference between high and low-depletion years. It is simply the difference between the mean of the overall springtime ozone between 300-200 hPa for both the LD and HD years, or do you somewhat average out the mean values for every pressure level, as shown in Fig. 5? How are then the values mentioned in lines 280-282 calculated? Please specify! From Fig. 5, I would assume that the ozone differences between LD and HD years are nowhere significant, so I don't understand quite well how the values, and their uncertainties, in lines 280-282 have been obtained.

**Answer**: Instead of interannual variability, we now explicitly use *"difference between high and low ozone depletion years"*.

We specified in the text how we calculate the differences: "*To provide a quantitative magnitude of these differences, we averaged the medians between 300-200 hPa (shown in Figure 5) and then calculated the ozone difference (and standard deviation) between low and high depletion years.*"

- In Fig. 6: can't the comparisons (vertical dashed lines) with the ozone medians in the NH obtained from IAGOS added to this figure, similarly as has been done for the SH?

**Answer**: Yes, we did it in a new figure (see below).

[Figure]

- Related to previous comment: the analysis of this figure 6 is only used for comparing the SouthTRAC measurements (2019 only) with the CAMS reanalysis output (the low-depletion years 2002 and 2019?). Is this the most important message from this Figure? Shouldn't the (pattern) agreement between Fig. 5 (observations) and Fig. 6 (model) be discussed first? E.g. the higher NH ozone values, the slightly (but not significant) higher amounts for LD years compared to HD years? And how do the model values relate with the measurement values (for all measurement types, and in NH and SH)?

**Answer**: We made some changes to the text to address the points mentioned: "*Figure 6 shows that the reanalysis reproduces the ozone interhemispheric differences, i.e., higher ozone in the NH and differences between high and low ozone depletion years. We also compared the pattern agreement between the ozone boxplots (CAMSRA) and in situ measurements. Figure 6 clearly illustrates the similarity between the ozone medians obtained from the CAMS reanalysis and in situ measurements (SouthTRAC and ozonesondes) particularly in the period with the highest number of flights, i.e., late winter-early spring, for pressures lower than 270 hPa. The lower agreement between the ozone medians of CAMS and in situ measurements in mid-spring period was determined by the lower flight frequency in this period, i.e., six flights during the mid-spring period versus ten in the late winter-early spring period (also discussed in section 4.2).*

- Lines 314-325: those findings are really not obvious at all from the figure. Are you saying that the agreement between CAMS and SouthTRAC (SH LD, light blue) is better during late winter –early spring compared to mid spring? On which ground? For pressures lower than 270 hPa (instead of below 270 hPa)? What do you mean with higher "fluctuations" in the medians for the mid-spring period? In the vertical or looking at the percentiles of the medians (I don't see this for the latter)? Are fluctuations in the vertical relevant given the coarser vertical pressure grid for CAMS?

**Answer**: Yes, that is our interpretation based on the ozone medians. Note that we replaced the term "below" with "lower than". Regarding the term "fluctuation," we modified the text: "*The lower agreement between the ozone medians of CAMS and in situ*

*measurements in mid-spring period was determined by the lower flight frequency in this period, i.e., six flights during the mid-spring period versus ten in the late winter-early spring period (also discussed in section 4.2).*

- Figure 7: in the discussion of Fig. 7, it might be informative to give the overall mean UTLS ozone value for each considered case (NH LD, NH HD, SH LD, SH HD).

**Answer**: We added: "*Mean UTLS ozone during high and low ozone depletion years was 399 and 425 nmol mol$^{-1}$ in the NH, and 228 and 265 nmol mol$^{-1}$ in the SH.*"

- Conclusions: if you put so much weight on the analysis in section 4.1 (two paragraphs devoted to it in the conclusions), you should stress much more the importance of this analysis in the main text as well. As written before, for me, it really hampers the logical flow of the paper a bit in its present form.

**Answer**: We have taken the suggestions (mentioned in the above points) relating to section 4.1 in terms of logical flow and reinforcing key messages, as well as the transitional role of that section.

**Technical corrections**

- Line 39: remove the dot before OH

**Answer**: We would like to retain the dot that indicates it is a radical species

- Line 86: remove the brackets before and after Neu et al., 2014

**Answer**: We removed them.

- Line 86: remove "diminished"

**Answer**: It does not apply in the current version.

- Line 90: remove "from"

**Answer**: We removed it.

- Line 93: in situ measurementS. Futher: replace "air masses exhibiting stratospheric character" with "air masses of stratospheric origin"

**Answer**: We replaced them.

- Line 135: replace so-called initiative with the Tropospheric Ozone Assessment Report – Phase II (TOAR-II) Focus Working Group HEGIFTOM (Harmonization and Evaluation of Ground-Based Instruments for Free-Tropospheric Ozone Measurements).

**Answer**: We replaced it

- Line 161: an identical instead of the identical

**Answer**: We modified it.

- Line 184: proposed by Charlton and Polvani (2007)

**Answer**: We modified as suggested.

- Line 207: Section 4.1 instead of Section 3.2

**Answer**: We replaced 3.2 by 4.2.

- Line 237: give the pressure level instead of "above 9 km".

**Answer**: We made the change suggested.

- Line 266: and instead of y

**Answer**: We changed it.

- Line 305: add ozone between percentage and difference

**Answer**: We made the change suggested.

- Line 358: replace "characterized by" with

**Answer**: We made the change suggested.

**Referee #2**

I want to thank the authors for incorporating my (and other) reviewer suggestions, increasing the amount of data analyzed, and adding confidence to the results contained in this manuscript. It is a much improved final product and I have only a few final technical comments:

- Line 90: This should be "tracers", correct?

**Answer**: We replaced "traces" by "tracers"

- Line 193: "reverse the zonal wind at 60ºS"

**Answer**: We made the change

- Line 266: "300 and 280 hPa"

**Answer**: We made the change

- Figure 7 Caption: change "lower less" to "less than"

**Answer**: We made the change

- Bottom of Page 16: Please add "iii." and "iv." to the final two paragraphs/conclusions

**Answer**: We added just the "iii" since the last sentence is more general.